



# Controls on brGDGT distributions in the suspended particulate matter of the seasonally anoxic water column of Rotsee

**Fatemeh Ajallooeian** [a]**, Nathalie Dubois** [a, b]**, Sarah Nemiah Ladd** [c]**, Mark Alexander Lever** [d, e]**, Carsten Johnny Schubert** [d, f]**, Cindy De Jonge** [a]

a Geological Institute, Earth Science Department, ETH Zurich, Sonnegstrasse 5, 8092 Zurich, Switzerland

b Swiss Federal Institute of Aquatic Science and Technology, Eawag, Überlandstrasse 133, 8600 Dubendorf, Switzerland

c Department of Environmental Science, University of Basel, Bernoullistrasse 30, 4056 Basel, Switzerland

d Institute of Biogeochemistry and Pollutant Dynamics, ETH Zurich, Universtitätstrasse 16, 8092 Zürich, Switzerland

e Marine Science Institute, University of Texas at Austin, TX, 78373 Port Aransas, USA

f Swiss Federal Institute of Aquatic Science and Technology, Eawag, Seestrasse 79, 6047 Kastanienbaum, Switzerland

Correspondence to: Fatemeh Ajallooeian (fatemeh.ajallooeian@gmail.com)





**Abstract**

Developing reliable methods for quantifying past temperature changes is essential for understanding Earth's climate evolution and predicting future climatic shifts. The degree of methylation of branched tetraethers (MBT'$_{5ME}$), of branched glycerol dialkyl glycerol tetraethers (brGDGTs), a group of bacterial membrane lipids, has become a widely accepted tool for lacustrine paleothermometry. To allow this, an empirical calibration was developed, based on MBT'$_{5ME}$ values of surface sediments across large spatial scales. As these sediments integrate variability across several years to decades, the sensitivity of MBT'$_{5ME}$ to seasonal and short-term environmental changes in the water column remains underexplored. Here, we present a record of brGDGTs in suspended particulate matter (SPM) from a monomictic, eutrophic temperate lake (Rotsee, Switzerland) over a 10-month period, examining both core lipids and intact polar lipids. Rotsee offers an ideal setting for this study due to its strong seasonal variations in temperature, conductivity, and dissolved oxygen caused by summer warming and associated stratification. In the oxic epilimnion, a minor increase in MBT'$_{5ME}$ during stratified summer months was caused by a rise in brGDGT Ia concentration. A similar increase in concentration of 6-methyl brGDGTs indicates a sensitivity to water temperature. In the seasonally anoxic hypolimnion, MBT'$_{5ME}$ correlated with water pH rather than temperature, suggesting that water chemistry influences this ratio, complicating its use as a temperature proxy. The production of intact polar lipid (IPL) tetraethers was observed exclusively in the anoxic hypolimnion during stratification, confirming anoxia as a key trigger for IPL tetraether production. Surface sediment samples along a depth gradient have a distinct depth-dependent distribution. Sediments below the oxic water column showed lower MBT'$_{5ME}$ values, likely due to the sedimentary production of brGDGTs IIa and IIIa. Sediments from seasonally anoxic areas reflected average epilimnion SPM values, suggesting the deposition of epilimnion brGDGTs into the sediments. This study of brGDGTs in Rotsee SPM and sediments thus indicates that temperature, pH and oxygen concentration impact GDGT distribution, with significant implications for using MBT'$_{5ME}$ as a temperature proxy in sediments from stratified lakes.



## 1. *Introduction*

Understanding local climate variability provides a critical foundation for addressing the pressing climate challenges of the present and future (Kaufman et al., 2020), allowing geographically focused efforts to mitigate the effects of climate change. Bacterial membrane lipid biomarkers, such as glycerol dialkyl glycerol tetraethers (brGDGTs), have emerged as a promising tool for paleothermometry (Russell et al., 2018). Initial empirical quantitative calibrations have been developed, based on the temperature dependence of their distribution in the environment (e.g. Weijers et al., 2006). Initially described in peatlands (Sinninghe Damsté et al., 2000; Weijers et al., 2006), brGDGTs have since been found in soils (Weijers et al., 2007a), aquatic sediments (Weijers et al., 2007b; Peterse et al., 2009; Tierney and Russell, 2009), and freshwater suspended particulate matter (SPM) (Tierney et al., 2010; De Jonge et al., 2014a; Russell et al., 2018; Martínez-Sosa et al., 2020). Their structural diversity used in paleoclimate proxies (Supp. Fig. S1) includes variation in the degree of methylation (four to six branches), termed tetra-, penta-, and hexamethylated brGDGTs. Additionally, internal cyclization of the methyl branches can lead to the formation of one or two cyclopentyl moieties. Penta- and hexamethylated brGDGT compounds with the outer methyl branch(es) on α and/or ω5 are termed 5-methyl brGDGTs, while those with the outer methyl branch(es) on α and/or ω6 are referred to as 6-methyl brGDGTs (De Jonge et al., 2013). The methylation of branched tetraethers index (defined originally as MBT and MBT', now MBT'$_{5ME}$) and the cyclization ratio of branched tetraethers (defined originally as CBT, now CBT') have been correlated with air temperature and soil or lake water pH, respectively (Weijers et al., 2007b; Tierney and Russell., 2009; Peterse et al., 2012; De Jonge et al., 2014b; Russell et al., 2018; Martínez-Sosa et al., 2019; 2021). Similarly, the isomer ratio (IR) of brGDGTs, which expresses the relative abundance of 6-methyl penta- and hexamethylated brGDGTs compared to their 5-methyl counterparts, has been used as a proxy for soil or lake water pH (De Jonge et al., 2014b; Naafs et al., 2017; Russell et al., 2018; Halffman et al., 2022), and more recently lake water conductivity and salinity (Raberg et al., 2021; Wang et al., 2021, Kou et al., 2022). Although variation in these ratios is generally interpreted as a response to temperature and/or pH changes (Russell et al., 2018; Martínez-Sosa et al., 2021), studies of lake systems have shown that additional environmental changes can impact brGDGT distributions (including individual brGDGT compounds and brGDGT-based ratios (MBT'$_{5ME}$, CBT', IR)). For instance, dissolved oxygen concentrations (Colcord et al., 2017, Weber et al., 2018, Van Bree et al., 2020, Yao et al., 2020, Lattaud et al., 2021), seasonal changes in mixing regimes (Loomis et al., 2014a, Van Bree et al., 2020, Dearing Crampton-Flood et al., 2020), nutrient concentrations (Loomis et al., 2014a, Hu et al., 2016), pH (Weijers et al., 2007b), and alkalinity (Schoon et al., 2013) have been shown to impact brGDGT distributions. Additionally, as water column studies have shown that brGDGT concentrations increase under $O_2$ depletion, it is thought that brGDGTs are primarily produced in the anoxic portion of the hypolimnion (Bechtel et al., 2010; Blaga et al., 2011; Woltering et al., 2012; Buckles et al., 2014; Loomis et al., 2014b; Miller et al., 2018, Weber et al., 2018, Van Bree et al., 2020), although the hypolimnion generally only reflects spring temperature. Additionally, soil inputs into lakes can contribute brGDGTs to the aquatic system, potentially altering their distribution in the water column. This can complicate the interpretation of brGDGT signals in paleoclimate reconstructions, as the mixing of brGDGTs from soil and aquatic sources are distinct.



It has been proposed that the variability of brGDGT distributions is primarily driven by microbial community composition, based on environmental (Weber et al., 2018; De Jonge et al., 2019, 2021) and pure culture studies (Sinninghe Damsté et al., 2018). This could impact phenotypic adaptations of bacteria to temperature, a phenomenon known as "homeoviscous adaptation", modelled by Naafs et al. (2021) and also shown in pure cultures (Chen et al., 2022; Halamka et al., 2022). While previous environmental studies in soils (Peterse et al., 2010; De Jonge et al., 2019) and culture studies (Sinninghe Damsté et al., 2011, 2014, 2018; Chen et al., 2022; Halamka et al., 2022) have shown that Acidobacteria are potential producers of GDGTs, Acidobacteria are generally not abundant in lake systems (Weber et al., 2018; van Bree et al., 2020). Furthermore, of the 15 brGDGT compounds identified in soil and aquatic ecosystems, many have not been detected in bacterial pure cultures. Recent studies have discovered biosynthetic genes associated with potential GDGT-producing pathways in a wide range of bacterial phyla, including those beyond Acidobacteria (Sahonero-Canavesi et al., 2022; Zeng et al., 2022). This broadens the scope of potential GDGT producers in lacustrine environments. Specifically, when GDGTs are correlated with bacterial abundance in lakes, Acidobacteria are often not considered clear candidates for GDGT production (Parfenova et al., 2013; Dedysh and Sinninghe Damsté, 2018; Weber et al., 2018; Van Bree et al., 2020).

In contrast to soils, which show no variability in brGDGTs between seasons (Weijers et al., 2011; Naafs et al., 2017), brGDGT concentrations and distributions in lakes vary seasonally, with reported increases in brGDGT concentrations during spring and fall isothermal mixing (Loomis et al., 2014a; Miller et al., 2018). This can introduce a seasonal production bias (Loomis et al., 2014a; Miller et al., 2018), and it remains unclear whether this seasonal behavior is driven by changes in water temperature, water chemistry (e.g., dissolved oxygen), or bacterial community composition (Shade et al., 2007). To elucidate which of these variables best explains seasonal variations in brGDGT concentrations and distributions, we examined water column and surface sediment samples from Rotsee (Switzerland). This lake experiences strong seasonal changes that include hypolimnion water anoxia during summer stratification (Fig. 1), and brGDGT presence was previously reported in surface sediments of Rotsee (Naeher et al., 2014) and in experimental mesocosms using water samples from the lake (Ajallooeian et al., 2024).

## 2. *Materials and Methods*

### 2.1. Water column, surface sediment and soil sampling

Rotsee (47°21'05.8" N; 8°31'12.7" E) is a small subalpine lake with a surface area of 0.48 km$^2$ and maximum depth of 16 m (Naeher et al., 2014). The lake is eutrophic and monomictic, exhibiting annual thermal water column stratification during the warm season. During this stratification period, high rates of aerobic mineralization of phytoplankton-derived organic matter and absence of physical mixing of oxygenated epilimnion water lead to anoxia in the hypolimnion (Schubert et al., 2010; Naeher et al., 2014) (Fig. 1).



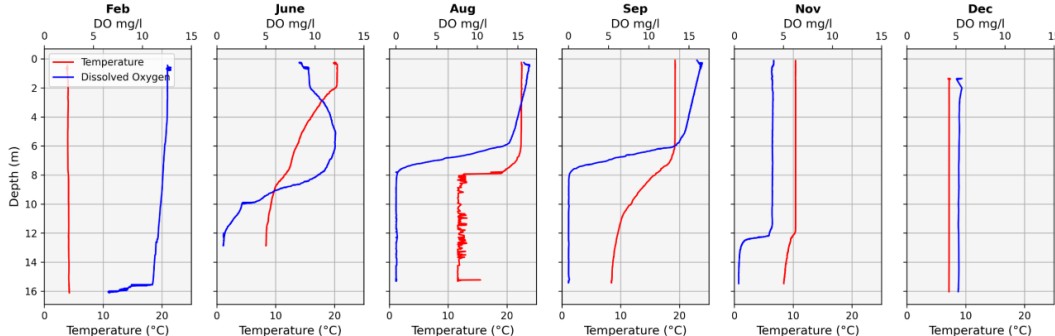

**Fig. 1.** Vertical profiles of temperature (°C, red) and dissolved oxygen (mg L$^{-1}$ blue) in the water column of Rotsee during selected months illustrating i) isothermal mixing (December and February), ii) stratification onset (June), iii) stratified water column (August and September) and iv) post-stratification conditions (November).

Starting in February 2019, water column samples were collected every two to four weeks. Water samples were taken using a 20 L Niskin water sampler at water depths of 0-1 m and 14-15 m (1 meter above the sediment surface), to represent the epilimnion and seasonally anoxic hypolimnion, respectively. Monthly time intervals until December 2019 (excluding April) were analyzed, resulting in a total of 10 time points. Throughout the water column, temperature, conductivity, pH, and dissolved oxygen were measured using a CTD scanner (Sea and Sun Technology®, Germany) at each timepoint. The mean annual air temperature (MAAT) was calculated based on the average measured air temperature of each month during the period of sampling (Feb-Dec 2019).

Water alkalinity was determined by analyzing aliquots from both depths using an 862 Compact Titrosampler (Metrohm Inc., Switzerland, EN ISO 9963-1:1995). From February to August, aliquots of the water were, moreover, used to measure concentration of anions (Nitrate – NO$_3^-$, Sulfate – SO$_4^{2-}$, Chloride – Cl$^-$), and cations (Calcium – Ca$^{2+}$, Sodium – Na$^+$, Ammonium – NH$_4^+$, Potassium – K$^+$, and Magnesium – Mg$^{2+}$) using a Compact Ion Chromatograph Pro, Model 881 (standard method from Metrohm Inc, Switzerland). Nutrients including total Phosphoros (total P) were measured on a Flow Injection Analyzer (SKALAR METHODS No. 461 (NO3/NO2/TN) and No. 503 (PO4/TP), instrument: SKALAR SAN++, Procon AG, Switzerland).

For each layer approximately 40 L of lake water was filtered within 12h to 24h after sampling and stored at 4 °C. Water was filtered using a 0.7 µm GF/F filter (Durapore®, Germany) placed on a titanium tripod (cleaned with EtOH and MilliQ between samples; referred to as GF/F sample) to collect suspended particulate matter (SPM). Subsequently, the filtered water underwent a second filtration step using a 0.22 µm PVDF filter (Durapore®, Germany) to capture smaller particles and free-living bacteria (referred to as PVDF sample). For some timepoints (17.07.2019, 14.08.2019, 18.12.2019), Aluminum Sulfate salt was added to lake water that was previously filtered over a 0.22 µm filter to coagulate dissolved organic matter (DOM). After coagulation, the resulting particles were collected by filtering through a 0.7 µm GF/F filter (referred to as DOM sample). This method has proven effective in flocculating DOM (Masion et al., 2000). Filters were wrapped in aluminum foil and stored frozen at -20℃.





To constrain the provenance of brGDGTs in SPM and the sediments, the dataset also includes four surface sediment
samples, collected along a depth transect. These samples were taken from 0-4 cm below the lake floor (blf) at depths
of 0.5 m (two samples), 5.5 m, and 11 m. Since the two 0.5 m samples showed very similar brGDGT distributions,
their average values are discussed. Furthermore, five soil samples from the surrounding watershed (4 gram sampled
from top 0-5 cm; Supp Table S1). The S0.5 surface sediment sample was collected from a shallow shoreline depth
(water depth: 0-5 cm, collection month: April), while S6 was collected from a water depth of 5.5 m that remains oxic
throughout the year (collection month: October). In contrast, S11 sample was obtained from a water depth of 11m,
that was anoxic at the time of sampling (collection month: September). The oxygen content of the sediment pore water
was not measured but is expected to be depleted within the top 0.5 cm at S6, and to be fully absent from sediments of
S11. Soil samples consisted of anthropogenic wetland, grassland, and forest soil samples immediately next to the lake.

**2.2. Lipid extraction**
From the GF/F filter samples that contained the bulk of the material, a subset of samples was selected for DNA
extraction (n= 20), where a known area of the filter (~16 mm$^2$) was stored for DNA analysis, before freeze-drying the
remaining filter material. The GF/F (n=20), a subset of PVDF (n= 6) and DOM filters (n= 2), surface sediment samples
(n=4) and soils samples (n=5) were freeze-dried before preparation for lipid extraction. Subsequently, filters were cut
into 3 equal sections, with split E1 of the water column samples extracted using a modified Bligh-Dyer extraction
(BDE+TCA) method with a mixture of methanol (MeOH), dichloromethane (DCM), and a phosphate-buffer (2:1:0.8,
v/v/v) for the first round of ultrasonic extractions (3x), and subsequently substitution of phosphate-buffer with 5%
trichloroacetic acid (TCA) for the second round of ultrasonic extractions (3x) (Sturt et al., 2004; modified from Pitcher
et al., 2009; Huguet et al., 2010). After either BDE or BDE+TCA extraction (3x), DCM was added, and this phase
was collected and dried under a gentle stream of N$_2$. The DCM phases were combined, providing the total lipid extract
(TLE). The second split of the filters (E2) were subjected to acid hydrolysis (AH) to convert all intact polar lipid (IPL)
GDGTs to core lipids (CL; after Weber et al., 2017). Briefly, the filters were placed in centrifuge tubes and submerged
in 1.5N HCl in MeOH (v/v). Tubes were capped and wrapped with Teflon tape and heated at 80℃ for 2 hours. A last
split (E3) was kept as an archive. Surface sediment samples were similarly extracted using modified BDE and
BDE+TCA methods.
Assuming a low relative abundance of IPLs in soils (e.g. De Jonge et al., 2019), we extracted brGDGTs (CL + unknown
contribution of IPLs) using an automated solvent extraction system (EDGE, ©CEM Corporations, USA) and
DCM/MeOH 1:1 (v/v) as the extraction solvent (3x).
The TLEs of SPM, surface sediment and soils were separated into fractions of different polarity using a Pasteur pipette
column packed with 3.5 cm of activated aluminum oxide, using three different solvent mixtures. The non-polar, ketone
and polar fractions were collected using hexane/DCM 9:1 (v/v), hexane/DCM 1:1 (v/v), and DCM/methanol 1:1 (v/v),
respectively. Before analysis, 49.6 ng of GTGT internal standard (C46) (Huguet et al., 2006) was added to the polar
fraction. The polar fraction was then filtered through a 0.45 μm PTFE filter, dried under N$_2$, and re-dissolved in 50 μL
of hexane/isopropanol (IPA) 99:1 (v/v). Subsequently, the samples were injected into a high-performance liquid



chromatography–mass spectrometry (HPLC–MS) system (Agilent Technologies®-1200, USA) as described in
Hopmans et al. (2016), using a modified column temperature of 40 °C and an injection volume of 10 μL.
Because of differences in ionization of the internal standard and brGDGTs (Huguet et al., 2006), the instrument error
in quantifying brGDGT concentrations was determined based on the repeated analysis of 12 freshwater column
samples, and estimated to be 15%, which has been used as an error estimate for the concentration of CL GDGTs
derived from E1. To calculate the quantity of IPL brGDGTs, the quantity of recovered brGDGTs from E1 (BDE+TCA:
CL brGDGTs) was subtracted from the E2 extracts (AH: CL+IPL brGDGTs) for both the water column filters and the
surface sediment samples. The instrument error in quantification was propagated for the IPL quantification (resulting
in an error of 17-21 %).

The MBT'$_{5ME}$ (De Jonge et al., 2014a) and Isomer Ratio (IR, De Jonge et al., 2014b) were calculated following the
formulas defined by De Jonge et al. (2014a; 2014b), where the IR reflects only compounds without cyclopentane
moieties (De Jonge et al., 2015; Halffman et al., 2022). The brGDGT-based reconstruction of water temperature and
pH was performed using the calibrations proposed by Russell et al. (2018).
$$MBT'_{5ME} = \frac{Ia+Ib+Ic}{Ia+Ib+Ic+IIa+IIb+IIc+IIIa}$$

$$IR = \frac{IIa\prime+IIIa\prime}{IIa\prime+IIIa\prime+IIa+IIIa}$$

$$DC' = \frac{Ib+IIb+IIb\prime}{Ia+IIa+IIa\prime+Ib+IIb+IIb\prime}$$

$$CBT' = \log_{10}\frac{(Ic+IIa\prime+IIb\prime+IIc\prime+IIIa\prime+IIIb\prime+IIIc\prime)}{Ia+IIa+IIIa}$$

Mean Annual Temperature (MAT) $= -1.21 + 32.42 \times MBT'_{5ME}$ ($r^2$= 0.92, p <0.0001, RMSE= 2.44 ºC)
Surface Water pH $= 8.95 + 2.65 \times CBT'$ ($r^2$= 0.57, p <0.0001, RMSE= 0.80)

**2.3. Quantification and sequencing of 16S rRNA genes**
To determine the bacterial community variability, a known fraction of the GF/F samples (~16 mm$^2$) was cut and stored
in PCR-clean tubes at -20℃ (n= 20). These samples underwent DNA extraction following the modular protocol
outlined by Lever et al. (2015), as done previously on Rotsee mesocosm SPM (Ajallooeian et al., 2024). To reduce
DNA adsorption, a 10 mM dNTP solution was added to the samples, followed by cell lysis solution I, and chemical
lysis treatment on a shaker for 1 hour at 50 °C to release DNA. The DNA-containing supernatant was then separated
from the residual sample material by centrifugation for 10 minutes at 14,000xg, washed twice with cold chloroform-
isoamyl alcohol (24:1) to remove non-polar fractions, and precipitated using NaCl, Linear Polyacrylamide (LPA; 20
μg mL$^{-1}$ of extract), and ethanol (EtOH) in a dark environment at room temperature for 2 hours. DNA pellets were
produced by centrifugation (20 mins at 14,000xg), washed three times using 70 % EtOH to remove excess NaCl, and



dried before resuspension and dissolution in molecular biology grade water ($H_2O$). QPCR standards consisted of
dilution series ($10^1$-$10^7$) of full-length 16S rRNA gene plasmids from *Rhodobacter sphaeroides*. As negative controls,
molecular biology grade $H_2O$ and extraction blanks were included. 16S rRNA gene copy numbers were >1,000-fold
lower in all negative controls compared to Rotsee DNA extracts.
Based on a subset of 18 samples (2 failed to amplify) that represent sampling dates throughout the year, a 16S rRNA
gene amplicon sequence library was prepared using the workflow outlined in Deng et al., (2020). In short, amplicons
of the bacterial 16S rRNA gene were obtained through PCR reactions using the primer pairs S-D-Bact-0341-b-S-17
(5′-CCTACGGGNGGCWGCAG-3′) and S-D-Bact-0785-a-A-21 (5′-GACTACHVGGGTATCTAATCC-3′). Paired-
end sequencing was performed using the Illumina MiSeq platform at ETH Zurich's Genetic Diversity Centre
(https://gdc.ethz.ch/). To ensure the quality and reliability of the sequencing run, an *Acidobacteria* positive control
(plasmids containing 16S rRNA gene sequences of *Holophaga foetida*) was included, along with contamination
controls consisting of molecular grade $H_2O$ and extraction blanks.
During the back-mapping process of the raw sequencing data, data loss was minimal (< 5%), with 14,608 zOTUs
(denoised sequencing data, zero radius operational taxonomic units (ZOTUs)) identified. After exclusion of singletons,
a total of 7,545,540 amplicon reads, representing 8501 ZOTUs, were used for further analyses, which included
operational taxonomic unit (OTU) clustering (97% identity threshold), and phylogenetic assignments using the SILVA
database (https://www.arb-silva.de/; further info in Deng et al., 2020). The resulting OTU table contained 8,299 taxa
across 18 samples. To avoid introducing biases based on differences in sequencing depths, the number of total reads
was rarefied to 222,646 reads per sample. This resulted in the retention of 6,103 OTUs and 16 samples for analysis.

### 2.4. Statistical methods

Mean ($\bar{x}$) and standard deviation ($\sigma$) of brGDGT fractional abundances and ratios for CL and IPL brGDGTs were
determined to examine variability through time, separately for epi- and hypolimnion water. As two compounds were
always below the detection limit (brGDGT IIIc and IIIc'), calculations were based on 13 brGDGTs. To calculate yearly
weighted averages for brGDGT fractional abundances and ratios, the average values for March and May were used to
represent the missing month of April in the dataset. Linear weighting was applied to months based on their normalized
concentrations, with higher weights being assigned to months with higher concentrations. Finally, concentration-
weighted averages of brGDGT ratios and fractional abundances were calculated, emphasizing on the influence of
months with greater lipid concentrations when determining the yearly average. To assess the extent to which
environmental variables account for variability in the brGDGT data, we conducted correlation analyses (Pearson
correlation coefficients (r)) based on concentration and fractional abundances and Principal Component Analysis
(PCA) based on standardized fractional abundances. Additionally, we calculated the variance in brGDGT fractional
abundance explained by each environmental variable while considering the effects of other variables via a stepwise
forward selection model (Dray et al., 2006, Legendre and Legendre, 2012, Russell et al., 2018). The stepwise forward
selection process constructs a linear regression model, starting with the environmental variable that exhibits the
strongest correlation ($R^2$) with the brGDGT data. Subsequently, it sequentially adds further variables based on the





significance of the F-statistic, determined through Monte Carlo permutation tests (499 simulations). The process
concludes when adding new variables no longer explains a significant fraction of the remaining variance, as
established through permutation testing (Legendre and Legendre, 2012).
The environmental and microbiome data were analyzed using packages, "phyloseq" (McMurdie and Holmes, 2013)
and "vegan" (Oksanen et al., 2013), implemented in R version 4.1.2. The rarefied bacterial communities at each depth
were aggregated based on taxonomic Order, and the Bray-Curtis dissimilarity method was employed to measure
differences in community composition in relation to brGDGT concentrations. A combined multivariate analysis of
variance (Adonis) was used to investigate whether the bacterial community in Rotsee potentially influences brGDGT
variability. This analysis was conducted using the Adonis function from the Vegan package (as used in Han et al.,
2020). The Adonis test was used to assess correlations between changes in CL and IPL brGDGTs (Ia, IIa, IIIa, IIa',
and IIIa') and microbial communities in the epilimnion and hypolimnion. IPL compounds below the detection limit
were assigned a value of zero. This analysis examined whether shifts in community composition were significantly
associated with increasing GDGT concentrations. OTU assignment to sample types (epilimnion/hypolimnion) was
based on an analysis that determined whether the occurrence of species in either the epilimnion or hypolimnion was
significantly higher than expected by chance, using 999 permutations and a significance threshold of $p < 0.05$. This
analysis identified a list of bio-indicator OTUs (Package indicspecies, De Cáceres, 2013). Adonis results are reported
in supplementary Table 3A and 3B. Additionally, the results for the separate bio-indicator test is reported in
supplementary Table 4A and 4B. Downstream and statistical data analysis were performed using the "scipy.stats"
package, from Python, with packages "matplotlib", "seaborn", "ternary" (python v.3.8.5) and "tidyverse", "ggplot2"
(from R v.4.2.3) used for data visualization and general data manipulation tasks.

## 3. Results

### 3.1. Mixing regime and water chemistry of Rotsee

In 2021, thermal stratification of Rotsee began in mid-May, with the thermocline and oxycline stabilizing at a depth
of 8 meters during July and August (Fig. 1). Following cooling of the epilimnion, water column mixing first deepened
the thermally stratified layer (September-November), with a fully mixed water column observed by December (Fig.
1). In the epilimnion of Rotsee, temperature varied significantly (4-24 °C), with February and August as the coldest
and warmest months, respectively. In the hypolimnion, August also was the warmest month, however, year-round
temperature variation was more limited (4-9 °C) (Fig. 2A). The seasonal mixing and development of a thermocline
caused variation in Rotsee's dissolved oxygen concentrations (Fig. 1). During the winter-spring mixing season,
dissolved oxygen levels in both the epilimnion and hypolimnion were stable, averaging around 11 mg L$^{-1}$. With the
onset and progression of thermal stratification, oxygen concentration in the epilimnion increased, reaching up to 15
mg L$^{-1}$ (Fig. 1). However, in the hypolimnion, oxygen levels began to decrease from May onwards, dropping to 1.57
mg L$^{-1}$, and resulting in suboxic conditions ([DO]< 2 mg L$^{-1}$) in May, June, and July, and anoxic conditions ([DO]<
0.1 mg L$^{-1}$) in August, September, and October (Fig. 1, Fig. 2E). The autumn mixing period facilitated the mixing of
oxygenated epilimnion water into the hypolimnion, resulting in a fully mixed water column by December.



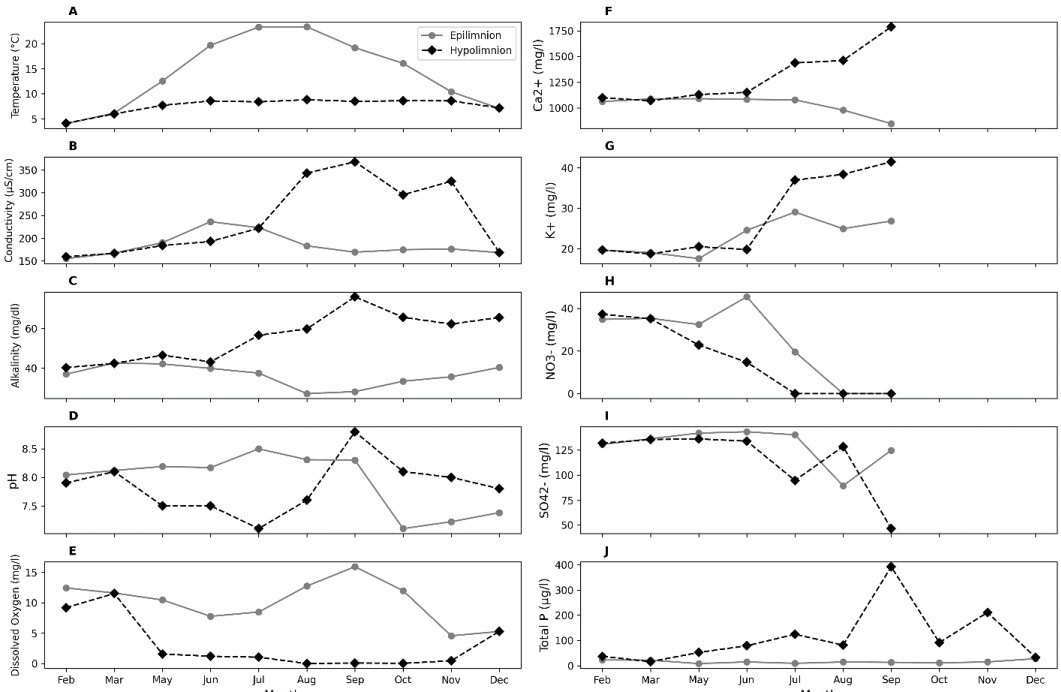


**Fig. 2.** The variability of inorganic parameters for epi- and hypolimnion of Lake Rot, with time. Specifically, temperature (°C),
conductivity (µS cm⁻¹), alkalinity (mg dl⁻¹), pH and dissolved oxygen (DO; mg L⁻¹), as well as cations ($Ca^{2+}$, $K^+$), anions ($NO_3^{2-}$,
$SO_4^{2-}$) (mg/ L⁻¹) and Total Phosphorus (Total P) concentrations (µg L⁻¹).

Seasonal stratification affected various inorganic chemistry parameters in the Rotsee water column. In the epilimnion,
these parameters are identified by a correlation with temperature, while in the hypolimnion, they correlate with
dissolved oxygen levels. Specifically, conductivity and total P showed significant correlations with temperature (r=
0.66, -0.70, respectively, p< 0.05) in the epilimnion, while in the hypolimnion, conductivity, temperature, and
alkalinity showed correlations with DO (r= -0.89, -0.70, -0.60, respectively, p< 0.05). $Ca^{2+}$ and $K^+$ followed the same
trend, correlating with hypolimnion conductivity (r= 0.91, p< 0.05 for both ions). Additional response to stratification
were observed for alkalinity, where hypolimnion alkalinity (Fig. 2C) increased with the onset of hypolimnion water
anoxia (Fig. 2C), while epilimnion alkalinity concentrations decreased during stratification (Fig. 2C). $Ca^{2+}$ (Fig. 2C)
correlated with alkalinity both in the epilimnion and hypolimnion (r= 0.85, 0.98, p< 0.05). Similarly, $K^+$, $SO_4^{2-}$, and
total P correlated with hypolimnion alkalinity (r= 0.92, -0.88, 0.69, p< 0.05, respectively) (Fig. 2F, G, J).

Water pH on the other hand displayed stratification-independent oscillations (Fig. 2D). In the epilimnion of Rotsee, it
showed stable values for the period of spring mixing and stratification (x̄= 8.1, σ= 0.1). With the start of the autumn's
isothermal mixing, pH dropped in the epilimnion to 7.1 (Fig. 2D). In the hypolimnion the stratification onset resulted
in reduced pH values (<7.1), increasing in pH (8.8) during the onset of isothermal mixing (Fig. 1).



To summarize the variance in the chemical parameters, a PCA ordination, based on water chemistry parameters
(dissolved oxygen, conductivity, pH, alkalinity, and cations and anions) of epi- and hypolimnion waters respectively,
was performed (Supp. Fig. S2). This ordination illustrates a similar water chemistry for spring and summer months
(March-June). September appears as an outlier in the ordination space of hypolimnion, driven mainly by changes in
$Cl^-$ and $SO_4^{2-}$ values (Supp. Fig. S2).

### 3.2. Patterns of brGDGTs in Rotsee suspended particulate matter (SPM)

#### 3.2.1. GDGT concentration variability

The PVDF and DOM samples yielded significantly lower summed concentrations in comparison to GF/F ($1 < \sum < 11$
ng $L^{-1}$) samples (Supp. Table S1) and often only included the brGDGT compounds Ia, IIa and IIIa. As GF/F filters
were thus able to collect 95% of brGDGT in the SPM, exclusively the results obtained from GF/F filters (Supp. Table
S1) will be discussed from this point on. The summed concentrations of CL brGDGTs varied between depths and over
time (Supp. Table S1; Fig. 3A). Generally, the hypolimnion had higher concentrations ($0.6$-$10.9 \pm 0.1$-$1.6$ ng $L^{-1}$) than
the epilimnion ($0.7$-$5.3 \pm 0.1$-$0.7$ ng $L^{-1}$) in the first half of the year. However, during the isothermal stratification
months (July, August, September, and October) and in December, the epilimnion concentrations exceeded those of the
hypolimnion (Fig. 3A). The IPL brGDGT concentration ranged from 0.1 to 2.4 ($\sigma = 0.7$) and from 0.02 to 3.6 ($\sigma = 1.2$)
ng $L^{-1}$ for epi- and hypolimnion, respectively (Supp. Table S1; Fig. 3A) with the hypolimnion IPL brGDGTs displaying
a notable increase in August (4 ng $L^{-1}$). The IPL brGDGTs generally comprised 15% of the total brGDGT pool in
epilimnion and 20% in hypolimnion throughout the year.
In the epilimnion, CL brGDGT Ia was the most abundant compound, with an increased concentration during the
summer months, resulting in a maximum concentration in July of 0.85 ng $L^{-1}$ (Fig. 3B). These elevated values for
brGDGT Ia persisted until August, after which a general decrease was observed, reaching 0.22 ng $L^{-1}$ in December.
The concentration of brGDGT IIa remained stable throughout the year, correlating positively with the concentration
of brGDGT Ia ($r = 0.67$, $p < 0.05$). In contrast, brGDGT IIIa was present at lower concentrations, with a stable average
of around 0.47 ng $L^{-1}$ until September. However, during the latter part of the year (October-December), it became
more abundant, with an average of 0.70 ng $L^{-1}$ (Fig. 3C). The opposing behavior between concentrations of brGDGT
Ia and IIIa (Supp. Fig. S3A), does not lead to a significant negative correlation. Compared with their 5-methyl
counterparts, 6-methyl brGDGTs consistently showed higher concentrations throughout the year in the Rotsee
epilimnion (Fig. 3B), with brGDGT IIa' generally present at a lower concentration than brGDGT IIIa'. The maximum
($0.94$ ng $L^{-1}$) and minimum ($0.26$ ng $L^{-1}$) concentrations of IIa' were observed in July and December, respectively,
mirroring the pattern observed for compound Ia (Fig. 3B). BrGDGT IIIa', with notably higher



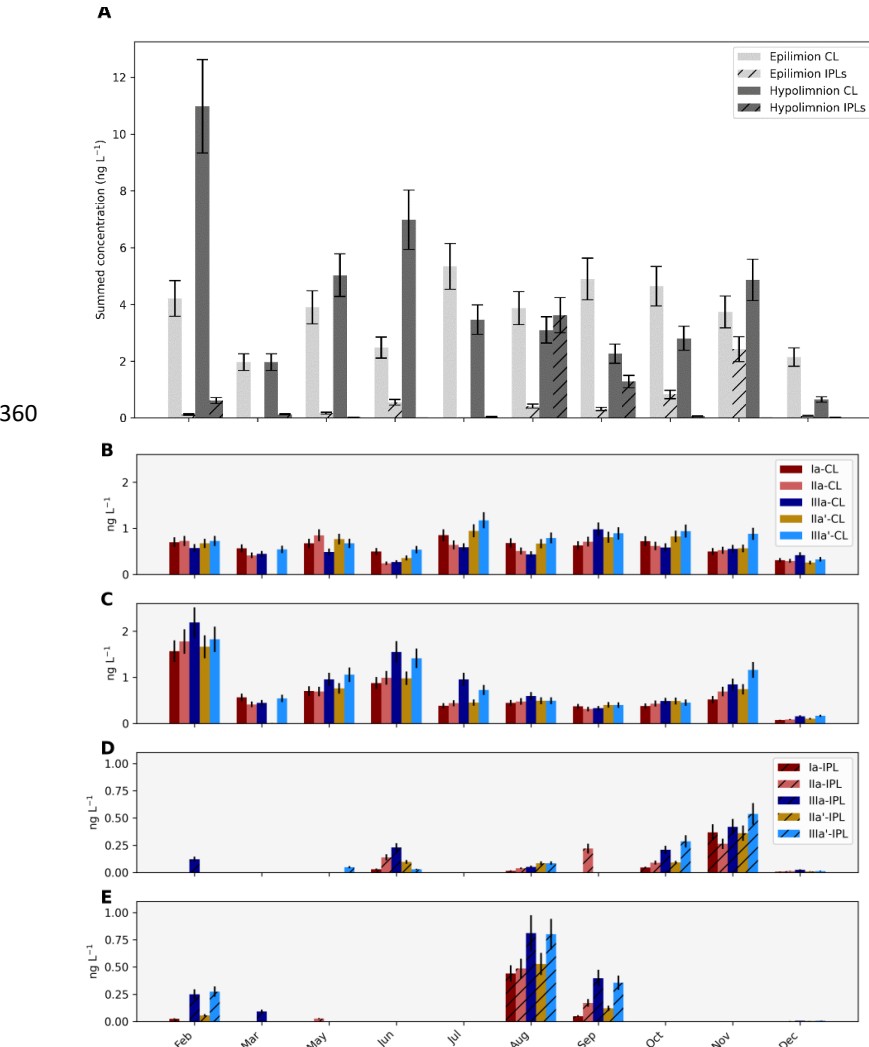

**Fig. 3.** A) Summed concentrations (in ng L$^{-1}$) of brGDGTs through the year in Lake Rot. Light grey bars represent epilimnion, while dark grey display hypolimnion concentrations. Subplots B-E display concentrations of the five most abundant brGDGTs, with B-D representing epilimnion values, while panels C and E represent hypolimnion concentrations. CL and IPL GDGTs refer to core and intact polar lipids, respectively. Error bars reflect the estimated instrumental error (15%).

concentrations, experienced a significant peak in July (1.17 ng L$^{-1}$) and maintained relatively high concentrations (0.79-0.94 ng L$^{-1}$) afterward, before declining to 0.37 ng L$^{-1}$ in December (Fig. 3B), also matching the concentration changes observed for brGDGT Ia. In the seasonally anoxic hypolimnion, brGDGT IIIa was the most abundant brGDGT compound, ranging from 0.1 to 2.1 ng L$^{-1}$ (Fig. 3C). In contrast to the epilimnion, a significant positive correlation was observed between the concentrations of brGDGT Ia and IIIa (r= 0.93, p= 0.00). Similarly, a strong positive correlation was observed between the concentrations of Ia and IIa (r= 0.98, p= 0.00) (Supp. Fig. S3A). The



6-methyl isomers (IIa', IIIa') did not reach the same concentration as observed in the epilimnion waters. Nevertheless,
brGDGT IIIa' remained one of the prevalent compounds, with an average concentration ($\bar{x}$) of 0.8 ng L$^{-1}$ ($\sigma$= 0.5 ng
L$^{-1}$, Fig. 3C).
For the IPL brGDGTs, only the predominant compounds (brGDGTs Ia, IIa, IIIa, IIa', and IIIa') were present above
detection limit, and that only during specific periods (Fig. 3D-E). These periods included June, August, October, and
November in the epilimnion, and February, August, and September in the hypolimnion (Fig. 3D-E). Notably,
hypolimnion IPL brGDGT concentrations exhibited exceptionally high values in August, reaching 3.62 ng L$^{-1}$. The
IPL form of brGDGT Ia was never the most abundant compound in either epilimnion (4-12%) or hypolimnion waters
(3-15%) (Supp. Table S1), which contrasts with the CL distribution. For the epilimnion, the summed 6-methyl
brGDGTs represent the largest fraction of IPL GDGTs (21-29%) while in the hypolimnion, the hexamethylated 5-
methyl GDGT (IIIa, 38%) represented the largest fraction of IPLs (Fig. 3D-E, Supp. Table S1).

### 3.2.2.   GDGT distribution variability

The seasonal changes in the concentration of CL brGDGTs in the epi- and hypolimnion of Rotsee result in
distributional changes that are summarized as variations in brGDGT ratios MBT'$_{5ME}$, IR and CBT', and MBT'$_{5ME}$-
based reconstructed temperatures (T$_{rec}$). In the epilimnion, MBT'$_{5ME}$ varied between 0.22-0.53, with a weighted
average value of 0.39 (Fig. 4A). The variation in MBT'$_{5ME}$ generally exhibited small changes from February to May
(0.38-0.39), caused by a stable fractional abundance of the major brGDGTs Ia, IIa and IIIa (Fig. 4C). In June, the CL
MBT'$_{5ME}$ showed a significant increase (T$_{rec}$: 15.8 °C), attributed to the high fractional abundance of brGDGTs Ia
(20%) and Ib (8%) (Supp. Table S1), which continued until August. In September, a drop in MBT'$_{5ME}$ value (0.31)
was coeval with an increased fractional abundance of brGDGTs IIb (> 6%) and IIIa (20%), while in December, where
MBT'$_{5ME}$ also declined (0.22), the lower fractional abundance of brGDGTs Ia (< 15%), Ib (< 4%), along with the
increased fractional abundance of IIIa (> 30%), contributed to this shift (Supp. Table S1). In the hypolimnion, the
range of MBT'$_{5ME}$ values (0.25-0.40) was narrower compared to the epilimnion (Fig. 4B). The MBT'$_{5ME}$ showed
maxima in March and September and decreased in June-July and Nov-Dec (Fig. 4B). While in March a high fractional
abundance in Ia was responsible for the elevated MBT'$_{5ME}$, in September the decreased fractional abundance of IIIa
along with a relative increase in Ib accounted for the increase in MBT'$_{5ME}$ value (Fig. 4D or Supp. Table S1). In July,
the low fractional abundance of brGDGT Ia and high fractional abundance of IIIa, drive the minimum MBT'$_{5ME}$ value
(< 0.25).
Reflecting the constant relative abundance of 5 and 6-methyl brGDGTs (brGDGT IIIa with and average value of 16%,
$\sigma$= 2% and brGDGT IIIa' with an average value of 20%, $\sigma$= 2%), the epilimnion showed low variability in IR values
($\bar{x}$= 0.56, $\sigma$= 0.06; Fig. 5A). In the hypolimnion (Fig. 5B), similar IR values compared to the epilimnion were observed.
March stands out with a noticeably low IR value (0.38) caused by a low (< 2%) fractional abundance of IIa'. In July
and September, variability in the IR values was caused by either an increased or decreased fractional abundance of
IIIa (30 and 14%, respectively). Reflecting the same variability in brGDGT diversity as the IR, the CBT' showed



constrained changes ($\bar{x}$= -0.03, -0.06; $\sigma$= 0.09. 0.04) in both the epi- and hypolimnion of Rotsee (Supp. Fig. S4).
Although DC' exhibited a slightly larger range in the epilimnion ($\bar{x}$= 0.25, $\sigma$= 0.05) compared to the hypolimnion ($\bar{x}$=
0.23, $\sigma$= 0.01, Supp. Fig. S4), the fractional abundances of compounds Ib, IIb, and IIb' remained similar across both
layers (Supp. Table S1).

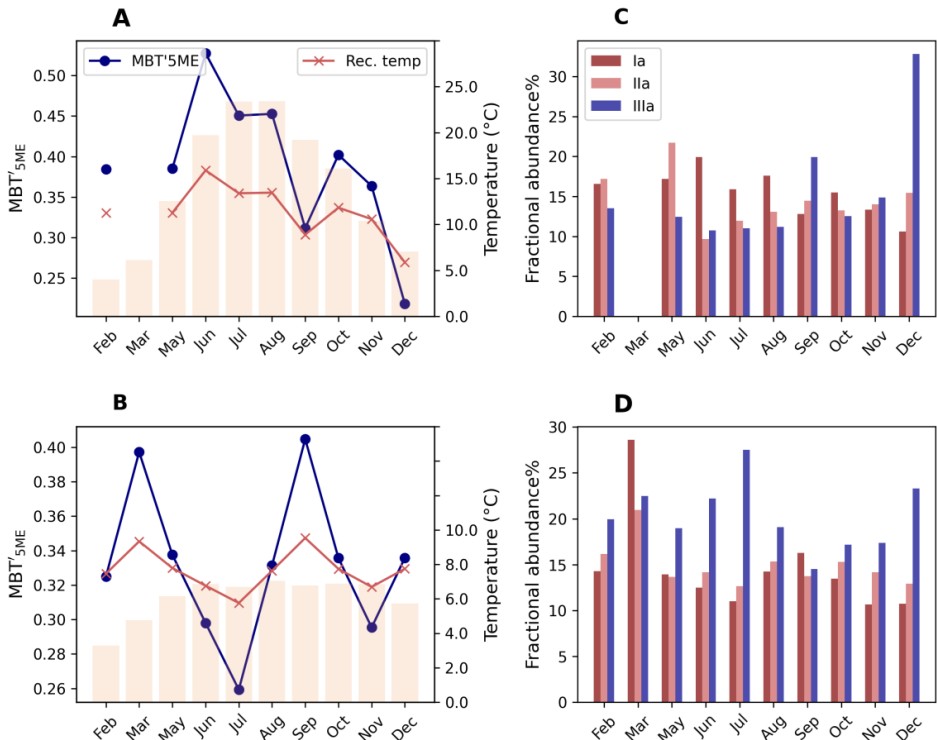


**Fig. 4.** Comparing brGDGT based ratio MBT'$_{5ME}$ values (blue) and the MBT'$_{5ME}$-based reconstructed temperature (red), with
measured water temperature at the depth of sampling (shaded bars). The fractional abundance of brGDGTs Ia, IIa and IIIa is plotted
in panels C and D. Panels A-C depict epilimnion values, panels B-D depict hypolimnion

**3.3. GDGTs of Rotsee surface sediments and surrounding soils**
In the surface sediment of Rotsee, the concentration of CL brGDGTs is similar in the two most surficial sediments
(0.5 and 6 m water depths), ranging from 159 to 203 ng g$^{-1}$ sed (Supp. Table S1). The deepest sediment, however,
shows a five-fold increase in concentration compared to the other sediments. The distribution of fractional abundance
of CL brGDGTs varies with depth, affecting brGDGT ratios MBT'$_{5ME}$ and IR (Fig. 6, Supp. Table S1).





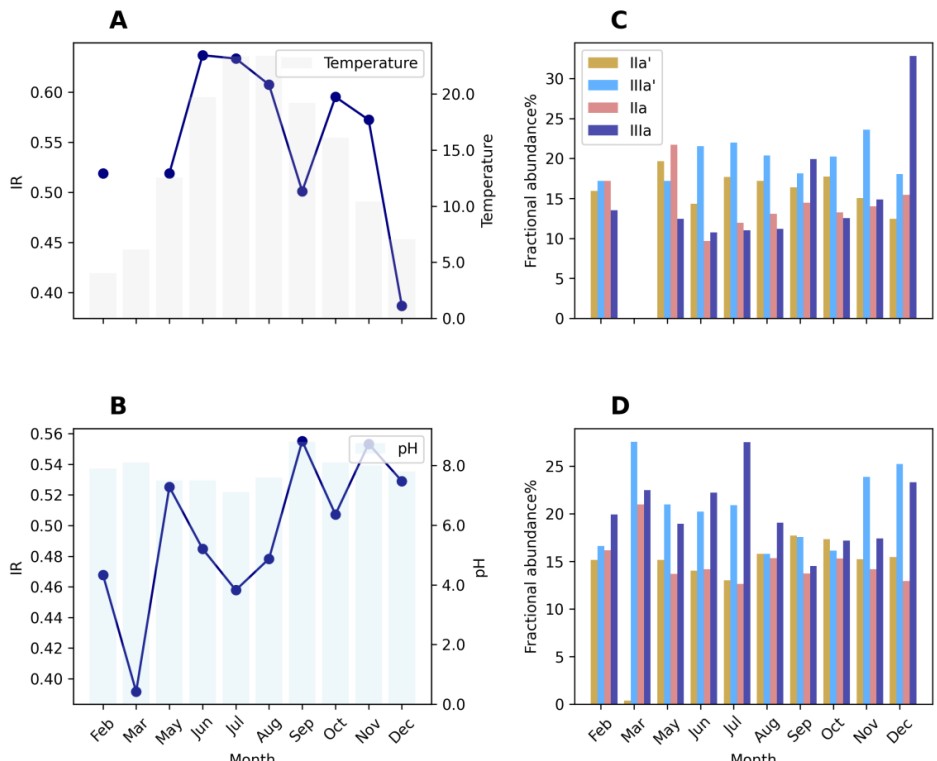

**Fig. 5.** Plotting brGDGT Isomer ratio (IR) values (blue)With measured epilimnion temperature (grey bars) pH plotted in (light blue). The fractional abundance of brGDGTs IIa, IIIa, IIa' and IIIa' is plotted in panels C and D. Panels A-C depict epilimnion values, panels B-D depict hypolimnion.

In the shallowest sediment (S0.5), the dominant CL compound is 6-methyl IIIa' (22%), while the sediment collected at intermediate depth (S6) shows a dominance of brGDGT IIa (19%) and IIa' (14%). Although the surface 6 m of sediments show a comparable fractional abundance of brGDGT Ia, IIIa, IIa' and IIIa' ($\bar{x}$= 13%, σ= 1%), the deepest sample (S11) shows more variability in the fractional abundance of other 5- and 6-methyl brGDGTs (Ia, IIIa, IIa', IIIa'). These differences result in a warmer MBT'$_{5ME}$ value for the shallowest sediment (MBT'$_{5ME = 0.40}$) compared to the sediment at intermediate depth (MBT'$_{5ME = 0.26}$). The intermediate sediments have a lower IR value (0.44) compared to shallow sediments (0.59), due to the high IIIa'% in shallow oxic sediment (Supp. Table S1). For the deepest sediment, 5-methyl brGDGT Ia is most abundant (17%), producing the warmest MBT'$_{5ME}$ signal of 0.43 (T$_{rec}$= 12.7 °C) of the sediment depth transect. The IR value in the deepest sediments is 0.55, similar to shallow oxic sediments, but due to an increase in IIa' GDGT (Fig. 6). The IPL-GDGTs in Rotsee surface sediment have lower concentrations compared to their CL counterparts, with the deepest sample having the highest IPL concentration (98.36 ng g$^{-1}$ sed), followed by the intermediate depth (44.50 ng g$^{-1}$ sed). Despite the concentration differences, the fractional



abundance pattern of these IPL compounds is similar across the complete depth transect (Fig. 6), with IPL-brGDGT
IIIa being the most abundant (> 25%), followed by IIIa' (> 15%).

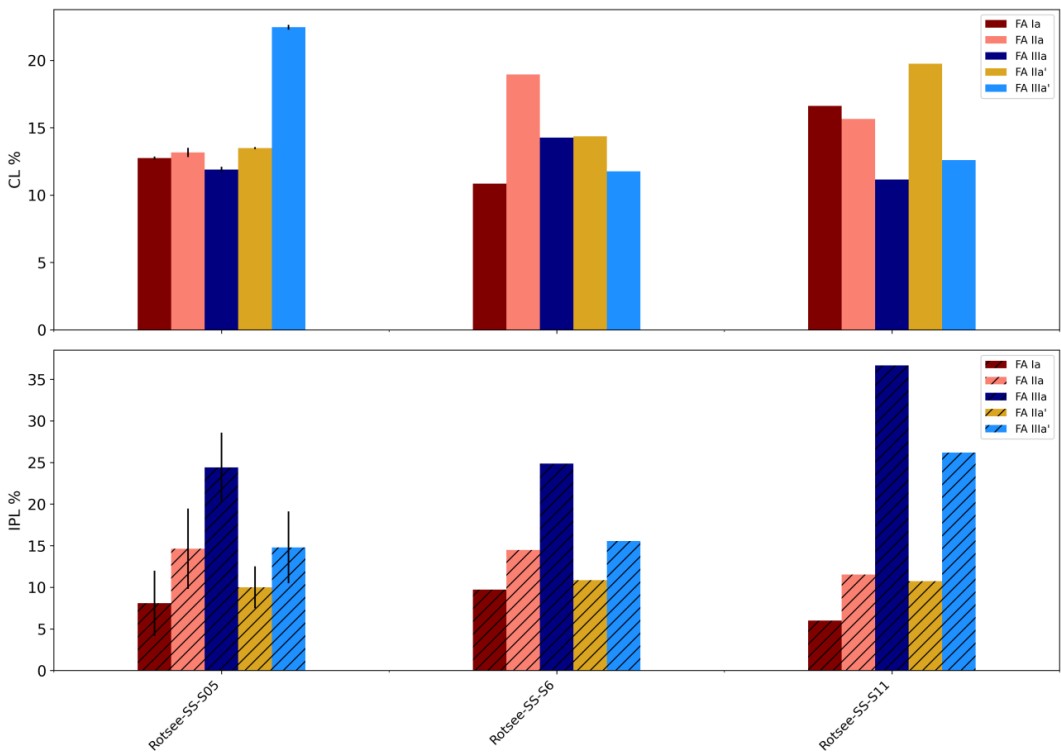


**Fig. 6.** The fractional abundances of CL and IPL brGDGTs in the surface sediment samples of Rotsee. The error bars represent the
standard deviation between the 2 shallow sediments collected for the S05 sample.
The surrounding soil samples of Rotsee (Supp. Table S1) showed varying brGDGT concentrations, with the highest
values encountered in wetland peat (2894 ng g$^{-1}$ soil), 10-fold higher than the northside grassland-forest soil (202.4
ng g$^{-1}$ soil). The brGDGTs fractional abundance in soil samples differs from the lake's surface sediment and water
column, with generally higher fractional abundance of brGDGT Ia (20-34%) and IIa (12-30%), resulting in a warmer
signal (MBT'$_{5ME}$ = 0.47-0.54). Additionally, soils around the lake exhibit generally lower IR values (0.17-0.19) and a
lower contribution of 6-methyl brGDGTs (generally <9%), demonstrating a different distribution compared to lake
water and sediments. A PCA is used to summarize changes in the brGDGT fractional abundance in Rotsee SPM,
surface sediments, and soil samples (Fig. 7A). As distinct brGDGT interdependencies are observed, separate
ordinations are also performed based on epilimnion and hypolimnion SPM respectively (Fig. 7B-C).



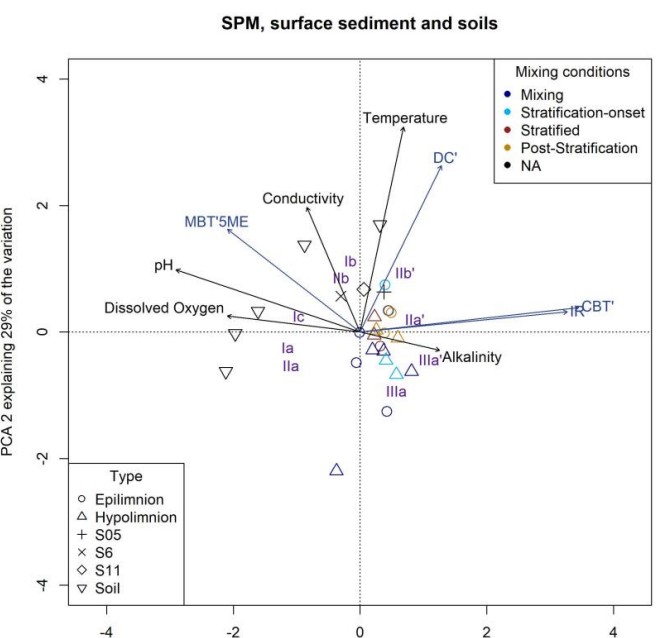

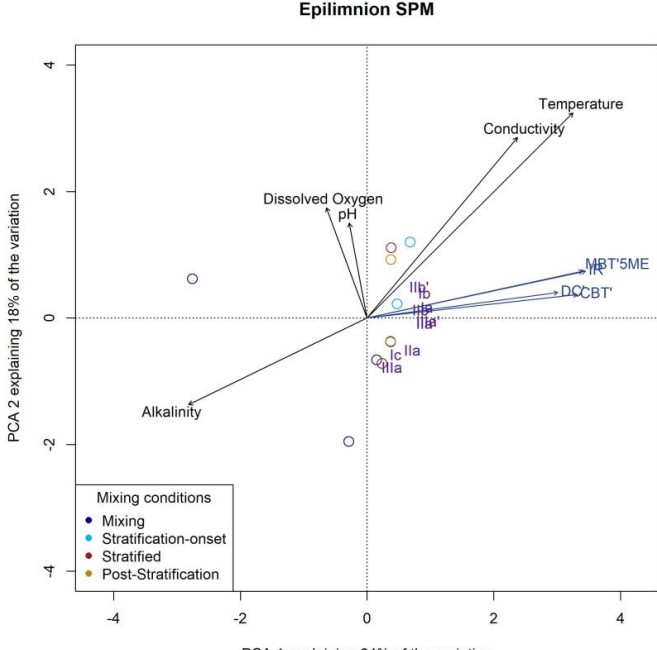

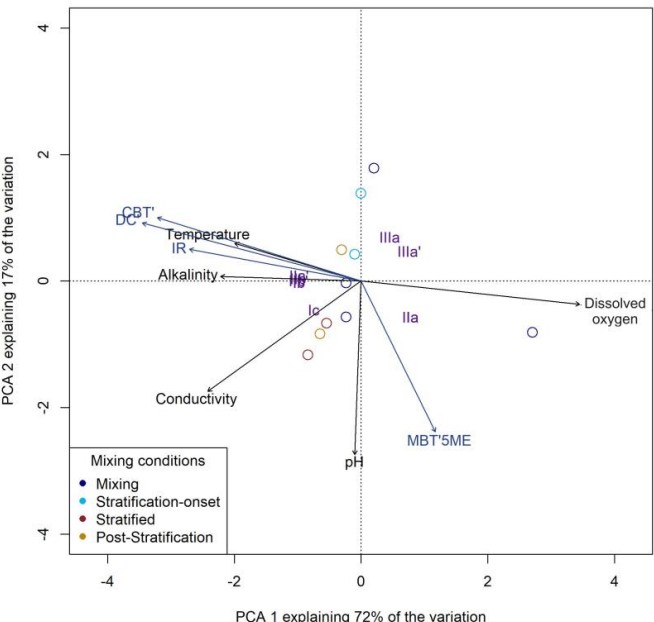

**Fig. 7.** An unconstrained Principal Component Analysis (PCA) based on the standardized fractional abundance of 11 CL brGDGTs (excluding IIIc and IIIc') in (A) Lake Rot suspended particulate matter (SPM), surface sediments and soils, (B) Lake Rot epilimnion SPM and (C) Lake Rot hypolimnion SPM. The symbol color code of the SPM samples reflects different mixing conditions. To improve readability only more abundant brGDGTs (IIa, Ib, Ic, IIa, IIb, IIIa, IIa', IIIa' and IIb') are plotted. The environmental variables (Temperature, alkalinity, conductivity, and dissolved oxygen) and GDGT-based ratios (MBT'$_{5ME}$, CBT', CBT', and DC') plotted a posteriori in the ordination space.


 

### 3.4. Rotsee 16S rRNA gene-based bacterial community

The qPCR-derived 16S mean gene copies (m L$^{-1}$) of the seasonal SPM exhibited concentration fluctuations throughout the year, showing increases and decreases in stratified and mixing months of both epi- and hypolimnion (Supp. Fig. S5A). The composition of the 16S rRNA gene-based bacterial community displayed seasonal variations, in both the epi- and hypolimnion (Fig. 8, Supp. Fig. S5B). During months with isothermal mixing the community composition was similar throughout the water column. The most significant contrast between the epilimnion and hypolimnion was observed during the stratified summer months (Fig. 8). Based on the permutational multivariate analysis of variance (Adonis), the concentration change of CL brGDGTs Ia and IIa in hypolimnion showed a significant correlation with the bacterial community composition. However, no such relationship was found in the epilimnion of Rotsee (Supp. Table S3A) or for the IPLs of epilimnion. A total of 63 Orders were significantly associated with either group (above or below the median CL-brGDGT Ia/IIa concentrations). And are reported in Supp. Table S3B. Additionally only 3 Orders were identified to show a concurrent marginal increase (P< 0.1) correlated with IPL-brGDGT IIIa' in the hypolimnion. These Orders were identified as WPS-2, Bacillales (Firmicutes) and Holophagales (Acidobacteriota) (Supp. Table S3B).

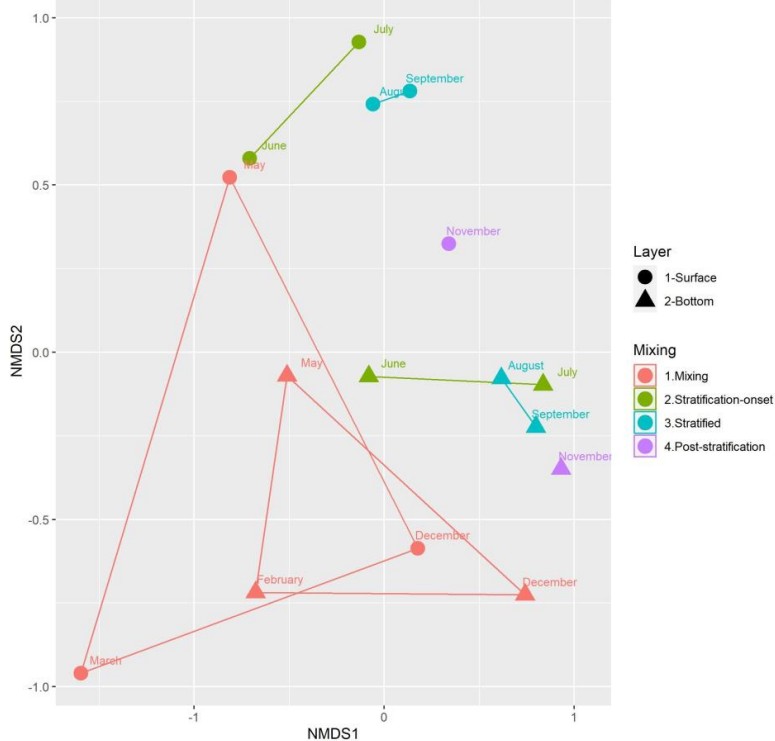

**Fig. 8.** Non-metric multidimensional scaling (NMDS) of Lake Rot 16S rRNA genes based bacterial community composition, with the shortest distance calculated as a polygon. The sample scores symbol color reflects the mixing condition of the water column during sampling, with symbol shapes representing sampling depth; epilimnion (sphere) or hypolimnion (triangle).





### 4. *Discussion*

#### 4.1. Abiotic and biotic drivers of brGDGTs production in the lake water column

In Rotsee, seasonal temperature changes and stratification impact brGDGT concentrations and distributions. Specifically, marked changes in brGDGT concentration are observed in the stratified summer months, during warming of the epilimnion and development of hypolimnion anoxia, coeval with changes in the bacterial community composition (Fig. 8).

When discussing the production of brGDGTs, IPL brGDGTs are often considered as markers of living (or recently living) GDGT-producing microbes that transform into more resistant CL GDGTs over time upon cell lysis (Lengger et al., 2013, 2014). In Rotsee, this process is observed for the conversion of hypolimnion August IPL brGDGT to epilimnion November CL brGDGTs. However, instances of increased CL brGDGTs concentrations that do not correspond to concurrent increases in IPLs are also observed, for instance, the increase of brGDGT Ia in epilimnion water. Conversely, there are instances of elevated IPLs that do not correspond to increased CL brGDGTs, for instance in Rotsee anoxic hypolimnion water. As both CL and IPL brGDGTs are produced in Rotsee, their production is therefore discussed separately.

Globally, brGDGT Ia is characterized by an increase in fractional abundance at warmer temperatures while brGDGT IIIa dominates the GDGT distribution in colder and/or deeper waters (Russell et al., 2018; Weber et al., 2018; Yao et al., 2020; Stefanescu et al., 2021). This temperature-sensitive production of CL is indeed evidenced from the increase in concentration of brGDGT Ia in the warmed and stratified summer months and brGDGT IIIa in colder mixing months in Rotsee. Also 6-methyl brGDGT IIIa' is apparently produced in the warmer summer months, which supports its interpretation as a marker for aquatic production (De Jonge et al., 2014; Guo et al., 2020; Ajallooeian et al., 2024). As there is no statistical significance correlating the concentration of CL-Ia with epilimnion bacterial OTU variability (Supp. Table S3A), bacterial community changes are not proposed to drive the temperature sensitive production of CL brGDGTs in the epilimnion.

The production of IPL brGDGTs in the hypolimnion is limited to anoxic conditions. This finding unequivocally highlights the role of anoxia as a key trigger for in-situ IPL brGDGT production. Culture studies have similarly reported the favorable production of brGDGTs (measured as CL GDGTs after hydrolysis) under oxygen-limited conditions (Chen et al., 2022; Halamka et al., 2022). BrGDGT IIIa and IIIa' dominate the distribution of IPL brGDGTs, hinting at the possibility that anoxic conditions could promote the production of hexamethylated brGDGTs. The increase in the concentration of brGDGT IIIa (and brGDGT IIIa'', a compound which was not observed in Rotsee) in suboxic to anoxic water columns have also previously been observed (Weber et al., 2018). However, this increase in IPL brGDGTs is not reflected in a corresponding rise in CL brGDGT concentrations. Moreover, the distribution of anoxic IPLs is distinct from that of the CL fraction (Fig. 3C and 3E), suggesting that the influence of anoxia on CL brGDGTs should be considered independently from their IPL counterparts. While the production of CL brGDGTs in the hypolimnion during periods of water mixing cannot be entirely ruled out, the CL brGDGT signal during these periods resembles that of the epilimnion. Nonetheless, the simultaneous increase in several bacterial OTUs in the hypolimnion along with rising CL brGDGT Ia concentrations indicates a potential link between the production of CL





brGDGTs and specific bacterial Orders. A different set of OTUs (Supp. Table S3B) showed a marginal correlation
with IPL brGDGT IIIa' in the hypolimnion, suggesting distinct sources or production mechanisms for CL versus IPL
brGDGTs in Rotsee's hypolimnion that could explain the distincy brGDGTs signal in the anoxic period of
hypolimnion.

### 4.2. Environmental drivers on brGDGTs concentration and distribution
### 4.2.1.  Proposed temperature-sensitive brGDGTs Ia, IIa and IIIa and ratios

In Rotsee, depth-dependent production of CL brGDGTs in the epi- and hypolimnion is thus observed, with distinct
dependencies of brGDGTs on environmental variables such as temperature, conductivity, alkalinity, pH, and dissolved
oxygen. Used as explanatory variables, they account for 86% of the variation in brGDGT distribution in the
hypolimnion and 67% in the epilimnion (Supp. Table S5). To understand the environmental drivers on brGDGTs, we
will separately discuss their impact on the epilimnion and hypolimnion.
BrGDGT Ia is typically interpreted to be produced in response to a temperature increase, as observed in globally
distributed lake sediments where its fractional abundance increases in surface sediments with warmer temperatures
(Russell et al., 2018; Martínez-Sosa et al., 2021; Raberg et al., 2021). In Rotsee, increased production of brGDGT Ia
during the warm summer months (Fig. 3B, Fig. 4C) is observed, supporting this interpretation. Additionally, the
concentration of brGDGT Ib, another compound known to increase with temperature in lake sediments globally (e.g.,
Raberg et al., 2021), shows a significant correlation with temperature ($r= 0.61$, $p< 0.05$). However, brGDGTs IIa and
IIIa don't show a direct correlation with temperature in terms of concentration and exhibit a negative relationship with
temperature in their fractional abundances (Supp. Table S2), which is also reflected in the PCA (Fig. 7B). Throughout
the remainder of the year (autumn and winter), as temperatures steadily decrease, the concentration of Ia generally
declines (Fig. 3B). This behavior aligns with the expected response of this compound to the cooling temperatures
typical of the colder months. However, in addition to a direct impact of temperature, the impact of lake water column
mixing needs to be considered. During the epilimnion mixing season, a decrease in brGDGT Ia and an increase in
brGDGT IIIa are observed, reflecting the GDGT distribution found in the hypolimnion (Fig. 3C). With the deepening
thermocline (October-November) and full water column mixing (November-December), hypolimnion brGDGT lipids
are brought to the epilimnion, supporting the potential role of specific anoxic bacteria as IPL GDGT sources.
Therefore, no direct impact of cooling on brGDGT Ia is observed in Rotsee. For brGDGT IIIa, in contrast, the increase
in concentration and fractional abundance (Fig. 3B, Fig. 4) during the colder November and December months in the
epilimnion is not derived from a hypolimnion water signal, indicating the cold-induced production of brGDGT IIIa.
However, although the increase in concentration of Ia is observed in warm stratified months in the epilimnion, the
absence of a correlation between Ia and temperature during colder months, contributes to the non-significant
dependency between MBT'$_{5ME}$ and temperature ($r= 0.59$, $p= 0.10$). In addition, MBT'$_{5ME}$ responds to the stratification-
dependent conductivity, showing a correlation of $r= 0.71$ ($p< 0.05$).



Although 6-methyl compounds are not traditionally associated with temperature sensitivity, an increased IIIa'
concentration, in response to warmer temperatures is notably visible in July (Fig. 3B). Furthermore, the negative
loadings of brGDGT IIIa' on epilimnion PCA axis 1 (Ia: -0.24, IIIa': -0.28) align with the loading of the temperature
vector (Fig. 7B). This temperature dependency of the fractional abundance of brGDGT IIIa' agrees with recent studies
(Russell et al., 2018; Martínez Sosa et al., 2020) that have observed positive correlations between the fractional
abundances of brGDGTs IIa' and IIIa' and growth temperature in aquatic environments. Interestingly, the IR, as evident
from Fig. 5A, demonstrates a more robust correlation with temperature in epilimnion waters ($r = 0.68$, $p < 0.05$),
compared with MBT'$_{5ME}$. Furthermore, the stepwise forward selection model confirms temperature as the primary
environmental variable, explaining 46% of the variance in IR in the lake's epilimnion (Supp. Table S4). The addition
of conductivity only marginally increases the explained variability by an extra 7% (resulting in a marginal effect
variance of 53%). This suggests that, while there is a significant linear correlation between IR and conductivity ($r = $
$0.65$, $p < 0.05$) that matches previous global observations (Raberg et al., 2021), temperature (with a 20 °C annual range)
may be the primary driver for variance in IR values, as supported by previous findings (Russell et al., 2018; Martínez-
Sosa et al., 2020; Ajallooeian et al., 2024). Nevertheless, the observed correlation between IR and conductivity further
indicates that in lakes where variation in conductivity is temperature-dependent, distinguishing the direct influences
of conductivity and/or temperature on IR can be challenging.
In the hypolimnion, a more muted variability in temperature (4-9 °C; Fig. 2A) is present. Hence, the indicated linear
correlations and stepwise forward selection models report a larger impact of water chemistry parameters on GDGTs
compared to temperature. Both the concentration of brGDGT Ia and IIIa' show a negative correlation with water
alkalinity ($r = -0.73$, $-0.69$, $p < 0.05$; Supp. Fig. S6). As alkalinity and temperature show a dependency ($r = 0.59$, $p < 0.1$)
the concentration of brGDGT Ia even displays a reverse correlation with temperature ($r = -0.66$, $p < 0.05$). Similarly,
the concentration of brGDGT IIIa, exhibits a correlation with water alkalinity ($r = -0.70$, $p < 0.05$) but not with
temperature. The lack of a temperature response in the hypolimnion water GDGTs can potentially be attributed to the
presence of distinct temperature dependent GDGT-producing bacteria in the epilimnion, and their absence in the
hypolimnion (Fig. 8). In the hypolimnion MBT'$_{5ME}$ shows a strong correlation with pH ($r = 0.80$, $p < 0.01$), highlighting
the various influences on this proxy in settings that do not experience a large temperature fluctuation.

**4.2.2.  Chemistry-sensitive 6-methyl and cyclopentane-containing brGDGTs, IR, CBT' and DC'**
Chemistry brGDGT ratios including CBT' and DC' do not exhibit any dependency on the water chemistry of the
epilimnion (Supp. Table S2), instead CBT' correlates with temperature ($r = 0.66$, $p < 0.05$). The absence of a correlation
between CBT' and pH particularly contrasts with previous lacustrine studies, where CBT' was found to correlate with
pH in oxic water layers (Zhang et al., 2016) and lake sediments (Martínez-Sosa et al., 2021). In Rotsee hypolimnion,
however, dissolved oxygen content (and conductivity and alkalinity to a lesser extent) seems to drive increases in
cyclopentane-containing and 6-methyl brGDGTs (Supp. Table S2). While the individual concentration of 5- and 6-
methyl brGDGTs do not exhibit a direct correlation with the dissolved oxygen levels, the fractional abundance of
cyclopentane containing brGDGTs IIb, Ib and Ic and 6-methyl brGDGTs IIa' and IIb' correlates with DO ($r = -0.64$ to




-0.80, p< 0.05), as well as the IR, CBT' and DC' (r= -0.65 to -0.78, p< 0.05). This correlation is also observed in the
IR (r=-0.64, r= 0.66, p< 0.05, for dissolved oxygen and alkalinity respectively). This aligns with findings from several
studies (Dang et al., 2018; Russell et al., 2018; Weber et al., 2018; Van Bree et al., 2020), which have reported a similar
anti-correlation between the fractional abundance of 6-methyl brGDGTs or IR and DO. On the other hand, other
studies (Yao et al., 2020; Qian et al., 2019) have argued for a positive relationship between 6-methyl compounds and
dissolved oxygen. Furthermore, as a correlation between alkalinity and total P was observed, IR correlates with total
P (r= 0.60, P< 0.05), suggesting that with increasing nutrients in the seasonally anoxic hypolimnion the proportion of
6-methyl brGDGTs also increased.

**4.3. Sedimentary brGDGT sources**
Prior to application of GDGT-based climate reconstructions based on sedimentary distribution, the contribution of
soil-derived brGDGTs needs to be constrained. The soil samples around Rotsee exhibit brGDGT distributions
significantly different from those in the lake's surface sediment (Fig. 9), as well as the water column.

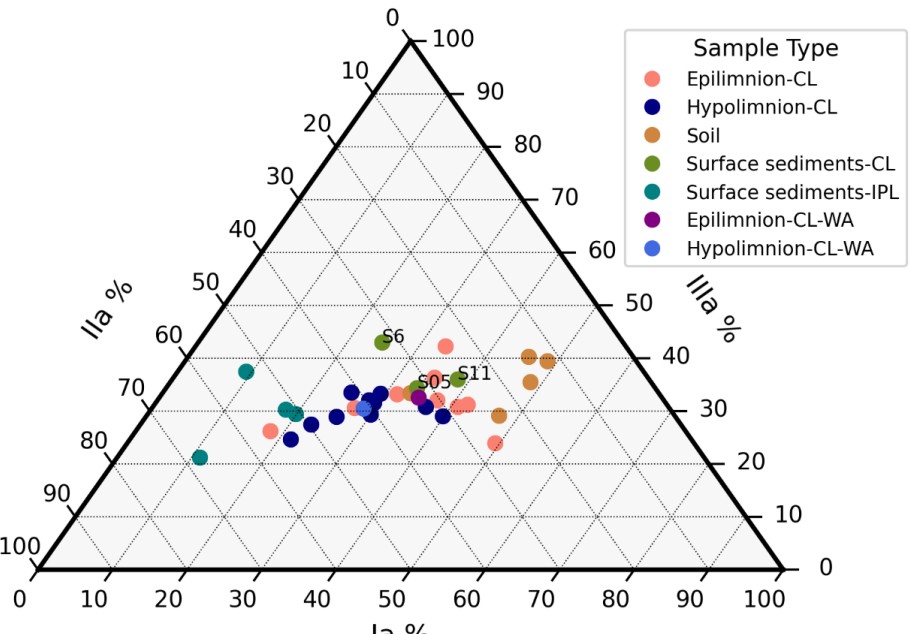


**Fig. 9.** Ternary plot based on fractional abundances of the brGDGTs Ia, IIa and IIIa, either core lipid (CL) or intact polar lipid (IPL)
brGDGT distributions. The sum of the fractional abundances amounts to 100%. Color is used to distinguish the SPM sampling
depth (epilimnion or hypolimnion, CL distribution), surface sediment (CL or IPL, the CL fractions are labeled in the plot.) and soil
samples. Weighted average of these fractional abundances of both epilimnion and hypolimnion are plotted in magenta and blue
(see legend).



The higher MBT'$_{5ME}$ and lower IR values in soils suggest minimal input of soil-derived brGDGTs into the lake.
Nevertheless, a distribution similar to the lake water column is observed in the grassland soil sample that are situated
close to the shallower part of the lake, suggesting soil input into waters around this area can not be completely
excluded. If sedimentary in-situ production would be absent, the lacustrine sediments would represent a mixture of
epi- and hypolimnion brGDGTs. However, while the shallow and intermediate sediments are expected to receive
brGDGTs dominantly produced in the epilimnion, a cold MBT'$_{5ME}$ signal of 0.27 is observed that is lower than the
epilimnion weighted average signal (MBT'$_{5ME}$= 0.40, 0.38 for stratified and mixed epilimnion, respectively). Across
the depth transect, the distribution of IPL and CL brGDGTs is unique compared to rest of the dataset (Fig. 9), which
suggests potential in-situ production of IPL and CL brGDGTs in the sediments. The in-situ production of penta- and
hexamethylated brGDGTs (IIa, IIIa specifically) might be influencing this MBT'$_{5ME}$ signal (Zhao et al., 2021 and
references therein). In the deepest sediment, the highest sedimentary brGDGT concentrations (Supp. Table S1) are
observed, which agrees with previous reports (Weber et al., 2018; Van Bree et al., 2020) on higher production of
brGDGTs under suboxic-anoxic conditions. On the other hand, the fractional abundances of Ia, IIa and IIIa in the
deepest sediment sample (11m depth) corresponds best with the average signal of the epilimnion (Fig. 9), leading to
a warmer MBT'$_{5ME}$ signal of 0.44 (T$_{rec}$= 13 °C). The MBT'$_{5ME}$ of the deepest sediments overestimate the water column
reconstructed mean annual temperature (MAT) by 2 degrees yet matches the current mean annual air temperature
(MAAT: 14 °C) at Rotsee. This suggests that the production and accumulation processes in the anoxic sediment are
different, potentially influenced by contributions from both epilimnion and hypolimnion SPM (Fig. 9).
The distinct differences in brGDGT distributions along the surface sediments depth transect highlight the importance
of coring location and lake water depth for interpreting MBT'$_{5ME}$-based temperature records. Considering these
distinct observations, the choice of coring location is crucial when intending to apply MBT'$_{5ME}$-based temperatures as
a paleotemperature indicator for lacustrine settings. While surface sediments could be preferred because of a stronger
contribution of epilimnion brGDGTs that show a good temperature dependency, in-situ production of CL brGDGTs
occurs at intermediate depths that are situated close to the chemocline. Deeper sediment samples that receive a mixture
of hypolimnion and epilimnion brGDGTs are not influenced as strongly by the production of CL brGDGTs.  As the
brGDGT interdependencies between epilimnion and hypolimnion are distinct, the shared temperature response
between the IR and the MBT'$_{5ME}$ results in a correlation (r= 0.86, p< 0.01) that can be used to identify GDGT
distributions that are sourced dominantly from the epilimnion. Furthermore, the interdependencies of brGDGTs
downcore can be compared with patterns observed in the epi- and hypolimnion (e.g. Fig. 7, Supp. Fig. 3B). As several
of the environmental dependencies we observe within Rotsee have been observed on a global scale, this approach has
potential to be used globally on distributed lakes with a thermally isolated hypolimnion. Its applicability through time
should however be tested in follow-up research.






## 5. Conclusions

In Rotsee, seasonal variability in temperature causes stratification, allowing to identify temperature, oxygen and pH as the most important environmental parameters affecting brGDGT distribution in the water column. Compared with the globally derived temperature dependency, MBT'$_{5ME}$ values show a muted response to water temperature in the epilimnion. The IR represents a stronger dependency on temperature highlighting the potential of using this proxy as a paleothermometer, although more extensive calibration work would be needed. Based on concentration changes of CL and IPL brGDGTs, production of both sets of compounds is observed. While CL brGDGTs are produced throughout the water column, the production of IPL brGDGTs seems confined to the anoxic hypolimnion. The significant production of CL brGDGTs in oxic environments can be expected to occur in a diversity of lakes. Although no bacterial groups in the epilimnion are identified to be significantly linked to GDGT production, a permutational multivariate analysis of variance identifies one Order of Acidobacterial OTUs and several non-Acidobacterial OTUs as potential CL-brGDGT producers in the hypolimnion, suggesting different producers in lakes compared to soils. Notably, a different group of OTUs, distinct from those associated with CL-GDGTs, including one Acidobacterial strain (Holophagales), are identified as potential IPL-brGDGT producers in the hypolimnion, indicating different sources of CL versus IPL producers of GDGTs in lakes.

The three surface sediments retrieved from 0-5 cm, 6 and 11 m depth transects of the lake put forward significant implications for paleotemperature reconstructions in lacustrine settings. Firstly, production of IPL-IIIa is uniformly observed for all sediments. When GDGTs are extracted using a high temperature extraction, the contribution of the IPL GDGT to the analysed GDGT pool will lower the reconstructed MBT'$_{5ME}$, Furthermore, CL GDGT production is observed in shallow sediments, especially, at the depth of the chemocline, which can complicate the interpretation of the MBT'$_{5ME}$ signal if a core is taken at this depth. However, in the deepest sediments that underlie the seasonally anoxic water column, the temperature signal of MBT$_{5ME}$ matches the epilimnion. Because of the depth location, a possible contribution of brGDGTs from the hypolimnion still can not be excluded. This suggests that paleotemperature studies based on brGDGTs recovered from cores collected from the deepest part of stratified lakes, a region usually targeted as sedimentation rate and bioturbation are minimal, may exhibit a muted temperature response. Based on the water column and sediments results, the authors suggest constraining the source of the brGDGTs within the water column by comparing MBT'$_{5ME}$ and IR in parallel. Potentially, this can be developed as a tool to recognize a dominantly epilimnion GDGT input in the sedimentary records of stratified lakes. This approach has potential to be used globally on distributed lakes with a thermally isolated hypolimnion.





*Data Availability*

The complete dataset for this work has been uploaded to the ETH Zurich research collection dataset under code
20.500.11850/696997.

*Author Contribution*

Fatemeh Ajallooeian was the main contributor, responsible for the conceptualization of the experiments, data curation,
formal analysis, visualization, and writing of the manuscript, including both the original draft and subsequent revisions
suggested by all co-authors. N. Dubois, S. N. Ladd, and C.J. Schubert provided critical feedback and input on the
manuscript. M. A. Lever supervised and contributed to the methodology for the microbiological aspects of the study.
C. De Jonge conceptualized the experiment, assisted with fieldwork, contributed to investigation and methodology on
biomarkers, and provided overall supervision, support, resources, and funding for the project.

*Acknowledgments*

This work was supported by the Swiss National Science Foundation [SNSF Project MiCoDy, grant PR00P2_179783].
In addition, the authors wish to express their gratitude for the assistance provided during the fieldwork of this project
by Patrick Kathriner, Karin Beck, Nina Studhalter, Sandra Schmid, Alois Zwyssig and for the valuable support
extended by the staff of the Genetic Diversity Center of ETH Zürich (GDC) in the laboratory work.

*Competing Interests*

The authors declare that they have no conflict of interest.



699

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

of intact polar glycerol dialkyl glycerol tetraethers from lacustrine suspended biomass. Limnology and Oceanography:
Methods, 15(9), 782-793.
Weber, Y., Sinninghe Damsté, J. S., Zopfi, J., De Jonge, C., Gilli, A., Schubert, C. J., Lepori, L., Lehmann, M. F.,
Niemann, H. (2018). Redox-dependent niche differentiation provides evidence for multiple bacterial sources of
glycerol tetraether lipids in lakes. Proceedings of the National Academy of Sciences, 115(43), 10926-10931.
Weijers, J. W., Schouten, S., Hopmans, E. C., Geenevasen, J. A., David, O. R., Coleman, J. M., ... and Sinninghe
Damsté, J. S. (2006). Membrane lipids of mesophilic anaerobic bacteria thriving in peats have typical archaeal
traits. Environmental Microbiology, 8(4), 648-657.
Weijers, J. W., Schouten, S., van den Donker, J. C., Hopmans, E. C., and Sinninghe Damsté, J. S. (2007a).
Environmental controls on bacterial tetraether membrane lipid distribution in soils. Geochimica et Cosmochimica
Acta, 71(3), 703-713.
Weijers, J. W., Schefuß, E., Schouten, S., and Sinninghe Damsté, J. S. (2007b). Coupled thermal and hydrological
evolution of tropical Africa over the last deglaciation. Science, 315(5819), 1701-1704.



Weijers, J. W., Bernhardt, B., Peterse, F., Werne, J. P., Dungait, J. A., Schouten, S., and Sinninghe Damsté, J. S. (2011).
Absence of seasonal patterns in MBT–CBT indices in mid-latitude soils. *Geochimica et Cosmochimica Acta*, *75*(11),
922 3179-3190.

Woltering, M., Werne, J. P., Kish, J. L., Hicks, R., Sinninghe Damsté, J. S., and Schouten, S. (2012). Vertical and
temporal variability in concentration and distribution of thaumarchaeotal tetraether lipids in Lake Superior and the
implications for the application of the TEX86 temperature proxy. Geochimica et Cosmochimica Acta, 87, 136-153.
Yang, H., Lü, X., Ding, W., Lei, Y., Dang, X., and Xie, S. (2015). The 6-methyl branched tetraethers significantly
affect the performance of the methylation index (MBT′) in soils from an altitudinal transect at Mount
Shennongjia. Organic Geochemistry, 82, 42-53.
Yao, Y., Zhao, J., Vachula, R. S., Werne, J. P., Wu, J., Song, X., and Huang, Y. (2020). Correlation between the ratio
of 5-methyl hexamethylated to pentamethylated branched GDGTs (HP5) and water depth reflects redox variations in
stratified lakes. Organic Geochemistry, 147, 104076.
Zang, J., Lei, Y., and Yang, H. (2018). Distribution of glycerol ethers in Turpan soils: implications for use of GDGT-
based proxies in hot and dry regions. Frontiers of Earth Science, 12, 862-876.
Zeng, Z., Chen, H., Yang, H., Chen, Y., Yang, W., Feng, X., Pei., H, and Welander, P. V. (2022). Identification of a
pRotseeein responsible for the synthesis of archaeal membrane-spanning GDGT lipids. Nature
communications, 13(1), 1545.
Zhang, Z., Smittenberg, R. H., and Bradley, R. S. (2016). GDGT distribution in a stratified lake and implications for
the application of TEX86 in paleoenvironmental reconstructions. Scientific reports, 6(1), 34465.
Zhao, B., Castañeda, I. S., Bradley, R. S., Salacup, J. M., de Wet, G. A., Daniels, W. C., and Schneider, T. (2021).
Development of an in situ branched GDGT calibration in Lake 578, southern Greenland. Organic Geochemistry, 152,
941 104168.
