# Peer review of "Controls on brGDGT distributions in the suspended particulate"

_EGUsphere, 2024_

## Author Comment (AC1)

Response to reviews for MS2-Control on **Controls on brGDGT distributions in the suspended particulate matter of the seasonally anoxic water column of Rotsee**

**Anonymous reviewer 2:**

Ajallooeian et al. presented a record of brGDGTs in suspended particulate matter (SPM) from a monomictic, eutrophic temperate lake (Rotsee, Switzerland) over a 10-month period, examining both core lipids and intact polar lipids, in addition to surface sediments and soils. The authors aimed to elucidate which environmental variables, such as water temperature, water chemistry (e.g., dissolved oxygen, pH, alkalinity, and conductivity), or bacterial community composition, best explain seasonal variations in brGDGT distributions, and thus examine the sensitivity of MBT'5ME to seasonal and short-term environmental changes in the water column. Overall, the study underscores the influences of temperature, pH, and oxygen on brGDGT distribution, raising important considerations for using MBT'5ME in temperature reconstructions from stratified lake sediments. The manuscript is well-organized, but several issues need to be addressed before acceptance.

As environmental factors influencing brGDGT production and distribution, temperature, conductivity, dissolved oxygen, pH, and alkalinity were considered in this study. Among these, the authors suggested that in the oxic epilimnion, MBT'5ME was associated with temperature, while in the seasonally anoxic hypolimnion, MBT'5ME correlated with water pH. However, considering Tables S2 and S5, it appears that in the epilimnion, MBT'5ME is related not only to temperature but also to conductivity. In the hypolimnion, MBT'5ME is influenced not only by pH but also by temperature. Does this imply that temperature, pH, and oxygen are not only complicating factors for the use of MBT'5ME, but also conductivity?

Thank you for your thoughtful feedback. In Rotsee, conductivity correlates with temperature in the epilimnion, likely due to the increased ion mobility at higher temperatures and potential evaporation effects during warmer months. This interrelation complicates the interpretation of $MBT'_{5ME}$, as temperature and conductivity may both influence brGDGT distributions. However, as shown in our data (Tables S2 and S5), the relationship between $MBT'_{5ME}$ and these factors is not straightforward, highlighting the complex interplay of multiple environmental drivers in Rotsee.

Conductivity can impact brGDGT distributions by influencing microbial activity and ionic strength in the water column, factors that may affect brGDGT production or preservation. Therefore, while the IR proxy strengthens the temperature-related interpretation for epilimnion-sourced brGDGTs, it is essential to consider the confounding influence of conductivity. Our findings emphasize the complexity of isolating temperature effects in lake systems where multiple environmental parameters are interrelated.

We have added lines 540 to 545 to clarify this point and emphasize the need for caution when interpreting $MBT'_{5ME}$ solely as a temperature proxy, particularly in stratified and eutrophic lakes where multiple environmental variables interact.

Added lines now read:

*"Additionally, MBT'$_{5ME}$ shows a significant correlation with stratification-dependent conductivity (r = 0.71, p < 0.05). This relationship likely reflects the interconnected effects of temperature on ion mobility and evaporation, which influence ionic strength in the water column. Conductivity may indirectly affect brGDGT production by altering microbial community structure or metabolic activity (Pearman et al., 2020) highlighting its role as a complicating factor for MBT'$_{5ME}$, especially in eutrophic and stratified lakes with tightly coupled environmental parameters."*

The authors proposed that the IR represents a stronger dependency on temperature in the epilimnion, highlighting the potential of using this proxy to identify brGDGT distributions dominantly sourced from the epilimnion within the water column by comparing MBT'5ME and IR in parallel. According to Table S2, IR also correlates with conductivity, similar to MBT'5ME. Would this suggest that although brGDGTs sourced from the epilimnion can be identified by comparing MBT'5ME and IR, MBT'5ME still does not fully reflect the influence of temperature alone?

Yes, you are correct—our data in Table S2 show that IR, like MBT'$_{5ME}$, also correlates with conductivity in the epilimnion. This suggests that, although comparing MBT'$_{5ME}$ and IR helps identify brGDGT distributions primarily sourced from the epilimnion, MBT'$_{5ME}$ potentially does not solely reflect temperature influences. The interplay between temperature and conductivity is now summarized in the manuscript in lines 540-545 as mentioned in the above reply.

Sediments from seasonally anoxic areas reflected average epilimnion SPM values, suggesting the deposition of epilimnion brGDGTs into the sediments. Does this suggest that the seasonal contribution from the anoxic hypolimnion plays a minor role in the application of MBT'5ME in such a lake? What could be the reason that the brGDGTs produced in the hypolimnion were not deposited or well-preserved in the sediments?

Thank you for highlighting this important point. Our findings indicate that sediments from the seasonally anoxic areas primarily reflect the average brGDGT composition from the epilimnion SPM, suggesting a relatively minor contribution from the hypolimnion. This could be due to several factors. (i) organic matter and brGDGTs produced in the hypolimnion may not settle effectively, as sedimentation dynamics favor epilimnion-derived inputs. (ii) organic matter from the hypolimnion could degrade during oxygenated phases of water column mixing, reducing its preservation in sediments (iii) the epilimnion, with its larger water volume and longer production period, likely dominates sedimentary inputs. (iv) the hypolimnion remains anoxic only seasonally, further limiting its overall contribution to the sedimentary record.

It is important to note that while these mechanisms are consistent with the data, further studies would be needed to definitively quantify the relative contributions of hypolimnion versus epilimnion-derived brGDGTs to lake sediments and we are only hypothesizing.

**Other comments:**

51-53: *"The production of intact polar lipid (IPL) tetraethers was observed exclusively in the anoxic hypolimnion during stratification, confirming anoxia as a key trigger for IPL tetraether production."* – If so, how should we interpret the IPL brGDGTs data shown in Figure 3D for the epilimnion?

We believe a potential explanation for the presence of IPL brGDGTs in the epilimnion (Figure 3D) lies in their seasonal distribution. The majority of IPL brGDGTs in the epilimnion are observed in October and November, while their production in the hypolimnion peaks during August and September. This temporal pattern suggests that IPL brGDGTs produced in the hypolimnion might be transported upwards during seasonal mixing events or deepening of the thermocline. This is highlighted in lines 496-502:

"The production of IPL brGDGTs in the hypolimnion is limited to anoxic conditions. This finding highlights the role of anoxia as a key trigger for in-situ IPL brGDGT production. While IPLs are detected in the epilimnion following the breakdown of stratification, i.e. during the mixing season (Fig. 3D), their distribution mirrors the hypolimnion pattern (Fig. 3C, D). This suggests that IPLs observed in the epilimnion are transported from the hypolimnion during the deepening of thermocline (October-November) and full water column mixing (November-December), therefore not supporting the production of IPL brGDGTs in the oxic epilimnion."

494-495: "*The production of IPL brGDGTs in the hypolimnion is limited to anoxic conditions. This finding unequivocally highlights the role of anoxia as a key trigger for in-situ IPL brGDGT production.*" – However, in Figure 1, bottom water anoxic conditions also occurred in June and November, yet IPL brGDGTs were not produced. What is the reason for this?

Thank you for your observation. It is important to distinguish between suboxic and anoxic conditions when interpreting our findings. Suboxic conditions refer to low but detectable levels of dissolved oxygen (e.g., periods like June or November in our water column), whereas anoxic conditions indicate the complete absence of oxygen e.g., July-October. These distinctions are critical, as the production of IPL brGDGTs appears to require fully anoxic environments, rather than suboxic ones. Please also refer to the answer in question above.

499: "*... brGDGT IIIa'', a compound which was not observed in Rotsee)*" – The compound brGDGT IIIa'' is mentioned here but is not shown in Figure S1, nor is there any explanation for it. Is it necessary to mention it here at all?

We have removed the mention of brGDGT IIIa'' from the manuscript as it is not observed in our data and does not contribute to our conclusions.

531: "*During the epilimnion mixing season, a decrease in brGDGT Ia and an increase in brGDGT IIIa are observed, reflecting the GDGT distribution found in the hypolimnion (Fig. 3C)*". – When is the mixing season in Figure 3? It would be helpful to add this information to Figure 3, as mentioned in Figure 1: (i) isothermal mixing (December and February), (ii) stratification onset (June), (iii) stratified water column (August and September), and (iv) post-stratification conditions (November).

We have added the seasonal phases (mixing, stratification onset, stratified, and post-stratification) to the caption of Figure 3 to clarify the corresponding periods and enhance interpretability.

[Figure]

**Fig. 3.** A) Summed concentrations (in ng L$^{-1}$) of brGDGTs through the year in Rotsee. Light grey bars represent epilimnion, while dark grey display hypolimnion concentrations. Subplots B-E display concentrations of the five most abundant brGDGTs, with B-D representing epilimnion values, while panels C and E represent hypolimnion concentrations. CL and IPL BrGDGTs refer to core and intact polar lipids, respectively. Error bars reflect the estimated instrumental error (15%). The water column dynamics of Rotsee can be categorized into four distinct phases: (i) isothermal mixing during December and February, (ii) the onset of stratification in June, (iii) a fully stratified water column in August and September, and (iv) post-stratification conditions in November.

537-539" "*However, although the increase in concentration of Ia is observed in warm stratified months in the epilimnion, the absence of a correlation between Ia and temperature during colder months, contributes to the non-significant dependency between MBT'$_{5ME}$ and temperature (r= 0.59, p= 0.10).*" – I guess in this text, Ia is referring to Ia-CL in Figure 3A. This should be clarified first.

Nonetheless, in my view, it is not clear in Figure 3A how the concentration of Ia increases during the warm, stratified months in the epilimnion.

We have clarified in the text that "Ia" refers specifically to **Ia-CL** shown in Figure 3A. We have slightly changed lines 537-539 to reflect the fractional abundance of Ia where the change in abundance is much more visible for the reader in figure 4.

Lines 537-540: "*However, although the increase in fractional abundance of CL-brGDGT Ia is observed in warm stratified months in the epilimnion (Fig. 4), the absence of a correlation between Ia and temperature during colder months, contributes to the non-significant dependency between MBT'5ME and temperature (r= 0.59, p= 0.10).*"

573-574: "*however, dissolved oxygen content (and conductivity and alkalinity to a lesser extent) seems to drive increases in cyclopentane-containing and 6-methyl brGDGTs (Supp. Table S2).*" – However, there was no relationship between IR and conductivity in Table S2.

Thank you for pointing out this inconsistency. We've updated the text to reflect that there is no **significant correlation** between IR and conductivity, aligning with the data in Supplementary Table S2.

577: "*... as well as the IR, CBT' and DC' (r= -0.65 to -0.78, p< 0.05).*" – delete IR in this sentence.

Noted and deleted.

577-588: "*This correlation is also observed in the IR (r=-0.64, r= 0.66, p< 0.05, for dissolved oxygen and alkalinity respectively).*" – r=-0.64 should be r=-0.64 in Table S2.

Noted and revised.

637-638: "*While CL brGDGTs are produced throughout the water column, the production of IPL brGDGTs seems confined to the anoxic hypolimnion.*" Does this mean that IPL brGDGTs are produced specifically in the anoxic hypolimnion, implying that they are not produced in the epilimnion or other oxygenated parts of the water column? If so, how should we interpret the IPL brGDGTs data shown in Figure 3D for the epilimnion? This point should be clarified to avoid any confusion.

See reply above to the question about the delivery of IPLs produced in the anoxic hypolimnion to the oxic epilimnion during full water column mixing.

In many places in the text, "GDGT" was used instead of "brGDGTs." It would be better to use "brGDGTs" consistently.

Noted and revised to brGDGTs throughout the text.

For "Rotsee" or "Lake Rot," it would be best to use one term consistently throughout the text and figure captions.

Noted and revised to Rotsee throughout the text.

Fig. 8. The layer legend (i.e., 1-Surface, 2-Bottom) can be replaced with 1-Epilimnion and 2-Hypolimnion to ensure consistency.

Noted and revised.

[Figure]

Fig. 8

Fig. 9: For figure, using different symbols for soil and surface sediment would be helpful.

Noted and revised.

[Figure]

Fig. 9

Newly added references to support added information in the manuscript:

Avila, M. P., Staehr, P. A., Barbosa, F. A., Chartone-Souza, E., and Nascimento, A. M. (2017). Seasonality of freshwater bacterioplankton diversity in two tropical shallow lakes from the Brazilian Atlantic Forest. *FEMS microbiology ecology*, *93*(1), fiw218.

Baxter, A. J., Peterse, F., Verschuren, D., Maitituerdi, A., Waldmann, N., and Sinninghe Damsté, J. S. (2024). Disentangling influences of climate variability and lake-system evolution on climate proxies derived from isoprenoid and branched glycerol dialkyl glycerol tetraethers (GDGTs): the 250 kyr Lake Chala record. *Biogeosciences*, *21*(11), 2877-2908.

Buckles, L. K., Weijers, J. W. H., Tran, X. M., Waldron, S., and Sinninghe Damsté, J. S. (2014). Provenance of tetraether membrane lipids in a large temperate lake (Loch Lomond, UK): implications for glycerol dialkyl glycerol tetraether (GDGT)-based palaeothermometry. *Biogeosciences*, *11*(19), 5539-5563.

LibreTexts. (2024). *Temperature effects on solubility*. LibreTexts Chemistry. Retrieved from https://chem.libretexts.org/Bookshelves/Physical_and_Theoretical_Chemistry_Textbook_Maps/Supplemental_Modules_%28Physical_and_Theoretical_Chemistry%29/Equilibria/Solubilty/Temperature_Effects_on_Solubility

Loomis, S. E., Russell, J. M., Ladd, B., Street-Perrott, F. A., and Sinninghe Damsté, J. S. (2012). Calibration and application of the branched GDGT temperature proxy on East African lake sediments. *Earth and Planetary Science Letters*, *357*, 277-288.

McManus, J., Collier, R. W., Chen, C. T. A., and Dymond, J. (1992). Physical properties of Crater Lake, Oregon: A method for the determination of a conductivity-and temperature-dependent expression for salinity. *Limnology and Oceanography*, *37*(1), 41-53.

Pearman, J. K., Biessy, L., Thomson-Laing, G., Waters, S., Vandergoes, M. J., Howarth, J. D., Rees, A., Moy, C., Pochon, A., and Wood, S. A. (2020). Local factors drive bacterial and microeukaryotic community composition in lake surface sediment collected across an altitudinal gradient. FEMS Microbiology Ecology, 96(6), fiaa070.

Ramos-Roman, M. J., De Jonge, C., Magyari, E., Veres, D., Ilvonen, L., Develle, A. L., and Seppä, H. (2022). Lipid biomarker (brGDGT)-and pollen-based reconstruction of temperature change during the Middle to Late Holocene transition in the Carpathians. *Global and Planetary Change*, *215*, 103859.

Watson, B. I., Williams, J. W., Russell, J. M., Jackson, S. T., Shane, L., and Lowell, T. V. (2018). Temperature variations in the southern Great Lakes during the last deglaciation: Comparison between pollen and GDGT proxies. *Quaternary Science Reviews*, *182*, 78-92.

Wu, J., Yang, H., Pancost, R. D., Naafs, B. D. A., Qian, S., Dang, X., Sun, H., Pei, H., Wang, R., Zhao, S., and Xie, S. (2021). Variations in dissolved O2 in a Chinese lake drive changes in microbial communities and impact sedimentary GDGT distributions. *Chemical Geology*, *579*, 120348.

Zhang, Z., Smittenberg, R. H., and Bradley, R. S. (2016). GDGT distribution in a stratified lake and implications for the application of TEX86 in paleoenvironmental reconstructions. *Scientific reports*, *6*(1), 34465.

---

## Author Comment (AC2)

Response to reviews for MS2-Control on **Controls on brGDGT distributions in the suspended particulate matter of the seasonally anoxic water column of Rotsee**

**Anonymous reviewer 1:**

The study of Ajallooeian et al. reports branched GDGT data and genetic data from SPM obtained from a large part of a single season in lake Rotsee, as well as comparisons with surface sediments and soils. The discussions follow the often-used strategy for interpretation of branched GDGT data, i.e. one correlates concentrations and indices with all environmental or genetic quantitative parameters which happened to be measured and discuss all significant, and some non-significant, correlations. As often happens, some of the correlations agree with previous studies and some do no leading to the often-made conclusion that multiple environmental parameters by influence brGDGT distributions.

The data are useful and add to a growing collection of such data which are not easy to obtain. The unfortunate thing is that I do not see (yet) how this data improved our understanding of lacustrine branched GDGT except 'it is complicated'. There have been more then plenty of those kind of studies on lake branched GDGTs (the far majority perhaps) so this is not a new insight. This is not the authors fault but one would like to see studies where more clarity is obtained and solid conclusions can be drawn. Can we still apply branched GDGTs as a lake temperature proxy? Should we just stop altogether in applying it? Is this single seasonal study of SPM sufficient to say that a brGDGT record from Rotsee is not useful for paleoreconstructions?

We appreciate the detailed and constructive feedback offered by this review. As the reviewer acknowledges, single study systems across one year have limitations, of which the most pertinent one is the difficulty in making a conclusive statement what this variability means for the downcore record. On the other hand, this type of study allows the temporal dynamics of GDGTs in a lake system to be described, which is a necessary step for understanding why a global MBT'$_{5ME}$ values are observed to depend on temperature. As such, this study reveals both mechanisms at play that determine the temperature dependency of the MBT'$_{5ME}$ (sections 4.1 and 4.2) as well as a section that discusses initial implications for the sedimentary record (4.3).

Specifically, our findings indicate that while MBT'$_{5ME}$ shows a response to temperature in the epilimnion, the onset of seasonal stratification plays a more significant role in the observed MBT'$_{5ME}$ shifts. This represents a novel insight, highlighting that MBT'$_{5ME}$ is not solely controlled by temperature but also by lake stratification dynamics.

Additionally, we discovered that in the hypolimnion, MBT'$_{5ME}$ responds more to chemical conditions than to temperature. These findings suggest that while MBT'$_{5ME}$ may not universally serve as a temperature proxy, its utility in paleoclimate reconstructions remains valid if the sedimentary brGDGT signal primarily reflects the epilimnion.

Importantly, this study shows for the first time that the Isomer Ratio (IR) shows a stronger correlation with mean annual temperature changes compared to MBT'$_{5ME}$ and the observed correlation between MBT'$_{5ME}$ and IR in epilimnion.

To address your concerns and clarify the role of stratification, we have updated sections 4 and 5, specifically adding and revising Lines 559-562 and 657-662. These revisions emphasize the novel contributions of our study, particularly regarding the stratification-related dynamics of MBT'$_{5ME}$ and the potential of IR as a reliable temperature proxy. The new lines now read:

Line 559-562 in section 4.2.1: "*The observed changes in MBT'$_{5ME}$ in the epilimnion are more indicative of stratification dynamics than direct temperature variability alone. During mixing events, the influx of hypolimnion-derived brGDGTs significantly alters the MBT'$_{5ME}$ signal, reducing its responsiveness to cooling temperatures in the epilimnion.*"

Line 657-662 in section 5: "*In Rotsee, seasonal temperature changes drive stratification, identifying temperature, oxygen, and pH as the most influential environmental parameters affecting brGDGT distribution in the water column. MBT'$_{5ME}$ values exhibit a muted response to water temperature in the epilimnion, however reflecting its sensitivity to stratification onset rather than a direct temperature dependency. In contrast, the IR demonstrates a stronger linear correlation with temperature changes, highlighting its potential as a reliable paleothermometer. While the application of IR for paleotemperature reconstructions shows promise, further calibration across diverse lake systems is necessary to establish its robustness.*"

My main criticism of the study (details below) is that it fully relies on correlations. For this to work, the quantitation, number of data points and representativeness of the samples must be sufficient. I am worried about these aspects. For example, as far as I can see there has not been a 'true' replicate analysis done (i.e. independent work-up of 2 parts of the same filter with addition of IS to the raw extract) so there is no way of knowing the real error in the quantitation. Some of the changes in concentration or indices are fairly small and we have no way of knowing if they are real changes or not. The number of data points is reasonable (12, but seems less for e.g. nutrients, i.e. 7 if I counted correctly) but I am unclear how representative is this season for Rotsee? Would another season have shown the same? Several studies have shown that biomarker lipid patterns can vary from season to season. I realize this is additional effort but it is a concern which needs to be addressed.

We appreciate the reviewer's concerns regarding the reliance on correlations and the robustness of our quantification. brGDGTs were "semi-quantified" following well-established protocols commonly used in the GDGT research community (lines 207-211). We ensured comparability among samples by using the entire TLE amount for each analysis and maintaining consistent sample preparation methods.

To address concerns about quantification variability, six independent samples underwent separate extraction processes, showing a concentration variation of 15-20%.

We also acknowledge the issue of data completeness. Due to logistical constraints, cation and anion measurements were unavailable for the last four months. Since no significant correlations were found between nutrient levels and GDGT distributions, these nutrient data have been moved to the supplementary material, and Figure 2 now includes only parameters with full coverage: conductivity, alkalinity, dissolved oxygen, temperature, and pH.

We will address each of the reviewer's concerns in detail in their comments below.

Detailed comments:

Line 40-41. ....has become a widely accepted tool for lacustrine paleothermometry.... Unsure about that wide acceptance for lakes but at least nothing is mentioned in the introduction on 'success stories' of this proxy in lake temperature reconstructions. Perhaps a few examples of apparently nice temperature records, as well as clearly wrong/biased records, would be useful in the introduction.

We appreciate the suggestion to provide context on the success and limitations of brGDGT-based temperature reconstructions in lakes. To address this, we have added references (Loomis et al., 2012; Watson et al., 2018; Ramos-Roman et al., 2022) to the introduction (lines 77-79), highlighting studies where brGDGTs have been successfully applied to lacustrine settings around the globe for temperature reconstruction. These examples illustrate both the potential and challenges of using GDGTs in different lake environments, providing a balanced perspective on their reliability and limitations. This addition clarifies the broader applicability of brGDGT proxies and sets the stage for our investigation into their specific responses in Rotsee.

New added/edited lines:

Lines 77-79: "*These calibrations have been successfully applied to generate high-resolution temperature reconstructions in lake sediment records, such as those from East Africa (Loomis et al., 2012), the Northern America (Watson et al., 2018), and the Mediterranean region (Ramos-Roman et al., 2022).*"

Lin 112. I would have expected some lake DNA (metagenome) studies as a reference.

Thank you for pointing this out. In Line 112, we reference studies that highlight the limited abundance of Acidobacteria in lake systems. Specifically, the studies by Weber et al. (2018), Van Bree et al. (2020), and Avila et al. (2017) demonstrate that Acidobacteria are generally less abundant in freshwater environments, including lake systems. For example, Weber et al. (2018) and Van Bree et al. (2020) explore European and African lake systems, where Acidobacteria are not major components of the microbial communities.

Additionally, on Line 117, we discuss studies by Dedysh and Sinninghe Damsté (2018) that suggest other microbial groups, beyond Acidobacteria, might be responsible for lacustrine GDGT production. These references, along with those by Parfenova et al. (2013), Weber et al. (2018), Van Bree et al. (2020), and Avila et al. (2017), support the argument that Acidobacteria are often not considered the primary producers of GDGTs in lakes.

The amplicon-based studies referenced, provide a robust basis for understanding microbial community composition in lake systems.

Line 124. Are these the only seasonal studies of lake brGDGTs? I thought there were more. If so, did they reach similar conclusions?

Thank you for your observation. You are correct that there are additional seasonal studies on lake brGDGTs. In response, I have included references to studies by Baxter et al. (2024), Zhang et al. (2016), and Buckles et al. (2014). These studies report varied findings, including a nonhomogeneous impact or lack of impact of seasonality, often with observed "cold" or "warm" biases on reconstructed MAT in their respective lakes.

Nevertheless, our findings align with the broader trends observed in these studies, which generally indicate that brGDGT distributions are influenced by multiple environmental factors beyond temperature alone. However, the specific responses can vary depending on the lake's characteristics and conditions. We have clarified this point and incorporated these references in the revised manuscript in lines 124-131:

"*In contrast to soils, which show no variability in brGDGTs between seasons (Weijers et al., 2011; Naafs et al., 2017), brGDGT concentrations and distributions in lakes vary seasonally, with reported increases in brGDGT concentrations during spring and fall isothermal mixing (Loomis et al., 2014a; Miller et al., 2018). This seasonality can introduce biases in reconstructed mean annual temperatures (MAT), resulting in "cold" or "warm" biases or even inconsistent seasonality effects, depending on the specific conditions and brGDGT production dynamics of the lake (Buckles et al., 2014; Zhang et al., 2016; Baxter et al., 2024). These findings underscore that brGDGT variability in lakes is often influenced by multiple factors beyond temperature alone, which can vary depending on the lake's unique characteristics*"

Line 206-209. This error is not the error in quantification as the ionization differences between IS and GDGTs were not corrected (I think? If so, the ng amounts calculated are just a complete guess) and the IS was added at the really final stage of fraction preparation (errors due to hydrolysis+workup are not included). At best this is a repeatability error of the instrument. I cannot therefore agree with the statement at these lines because this error really represents the (unrealistic) best case scenario and ignores the complete workup and sample inhomogeneity errors. I would recommend splitting a sediment sample, or even better a filter cut in two pieces, and work this up completely separately to obtain a better constraint of the overall quantification error.

Thank you for your detailed feedback on this aspect. We acknowledge the limitations in our current approach to error estimation, as the internal standard was indeed added at the final stage and does not account for errors from the entire workup process, including hydrolysis and extraction. We agree that this primarily reflects instrumental repeatability rather than a comprehensive quantification error. The variability in ionization efficiency is expected to be captured by the re-analysis of individual samples and thus by the instrument error as reported. In addition, we maintained consistent sample preparation protocols that were applied on very similar sample types (all filtered SPM).

As mentioned in our response to your first comment, we now also identified six independently processed samples where the entire extraction and workup steps were repeated. The observed variation in brGDGT concentrations between these samples was within 15-20%, i.e. the same as the previously reported instrument error. For example, in the July-epilimnion sample, brGDGT Ia concentrations ranged from 0.84 ng/L to 0.98 ng/L, while IIIa ranged from 1.17 ng/L to 1.33 ng/L. These details have been added to the methods section (lines 217-220). This data is now added to the supplementary Table 1 of the article as well.

Line 375-380 and line 483. IPLs are nearly always a small fraction of the total lipids. Since direct measurements of IPLs were not done, this could even be an overestimate, i.e. if some of the GDGTs

released by acid hydrolysis were not derived from IPLs but from matrix-bound material. Why are these IPL abundances so low, if the assumption is that all this material is derived from living biomass? Do branched GDGTs mostly occur as CL in the cell? Is cell lysis so quick or happening during filtration?

Thank you for raising these important points. Our data, along with observations from surface sediment records, indicate that GDGTs are indeed present in both core lipid (CL) and intact polar lipid (IPL) fractions, often displaying distinct distribution patterns. This suggests that GDGTs can be produced in both forms, potentially reflecting different stages of microbial activity or degradation processes.

We have highlighted these aspects in lines 491-498 referencing Raberg et al. (2022). Our data, along with observations from surface sediment records, indicate that GDGTs occur in both core lipid (CL) and intact polar lipid (IPL) fractions, potentially reflecting different stages of microbial activity or degradation processes. Raberg et al. (2022) observed that IPL brGDGTs constitute a small fraction of the total lipids generally. Particularly, IPL with phospholipid headgroups that are more abundant in water column, might degrade more readily into CLs compared to another group of IPLs.

We propose that degradation likely plays a role in the transition from IPLs to CLs, particularly for IPLs with phospholipid headgroups according to Raberg et al., (2022), but the relative contributions of cell lysis and in situ production to GDGT transformation remain unresolved.

New added/edited lines 491-498:

"*Our data, along with observations from surface sediment records, indicate that brGDGTs occur in both core lipid (CL) and intact polar lipid (IPL) fractions, often displaying distinct distribution patterns. This suggests that brGDGTs can be produced in both forms, potentially reflecting different stages of microbial activity or degradation processes. Raberg et al. (2022) observed that IPL brGDGTs generally constitute a small fraction of the total lipids. Particularly, IPLs with phospholipid headgroups, which are more abundant in the water column, might degrade more readily into CLs compared to other IPL types. It is proposed that degradation likely plays a role in the transition from IPLs to CLs, especially for IPLs with phospholipid headgroups, but the relative contributions of cell lysis and in situ production to brGDGT transformation remain unresolved.*"

Line 385 and further. Coming back to the 15% error in quantification. It is not indicated here what the errors in brGDGT indices are. Are the changes observed larger than the assumed errors? At least replicate analysis could be done but preferably replicate work up.

The error in brGDGTs indices is based on the duplicate measurements of 12 samples as mentioned in lines 212.

Fig. 2. There is a clear mismatch in the no. of data points with only 7 data points for the cat- and anion concentrations. Can we really make solid conclusions on so few datapoints?

We agree that drawing solid conclusions from a limited dataset may be problematic. To address this, we have updated Figure 2 to include only those inorganic parameters (conductivity, alkalinity, dissolved oxygen, temperature, and pH) for which data are available throughout the entire sampling period. Some nutrient values, which were available for a shorter duration, have been moved to the

supplementary data section. The manuscript has also been updated to reference the correct data in Figure 2 and the supplementary data for the nutrient values throughout.

Current figure 2:

[Figure]

Newly added references to support added information in the manuscript:

Avila, M. P., Staehr, P. A., Barbosa, F. A., Chartone-Souza, E., and Nascimento, A. M. (2017). Seasonality of freshwater bacterioplankton diversity in two tropical shallow lakes from the Brazilian Atlantic Forest. *FEMS microbiology ecology*, *93*(1), fiw218.

Baxter, A. J., Peterse, F., Verschuren, D., Maitituerdi, A., Waldmann, N., and Sinninghe Damsté, J. S. (2024). Disentangling influences of climate variability and lake-system evolution on climate proxies derived from isoprenoid and branched glycerol dialkyl glycerol tetraethers (GDGTs): the 250 kyr Lake Chala record. *Biogeosciences*, *21*(11), 2877-2908.

Buckles, L. K., Weijers, J. W. H., Tran, X. M., Waldron, S., and Sinninghe Damsté, J. S. (2014). Provenance of tetraether membrane lipids in a large temperate lake (Loch Lomond, UK): implications for glycerol dialkyl glycerol tetraether (GDGT)-based palaeothermometry. *Biogeosciences*, *11*(19), 5539-5563.

LibreTexts. (2024). *Temperature effects on solubility*. LibreTexts Chemistry. Retrieved from https://chem.libretexts.org/Bookshelves/Physical_and_Theoretical_Chemistry_Textbook_Maps/Supplemental_Modules_%28Physical_and_Theoretical_Chemistry%29/Equilibria/Solubilty/Temperature_Effects_on_Solubility

Loomis, S. E., Russell, J. M., Ladd, B., Street-Perrott, F. A., and Sinninghe Damsté, J. S. (2012). Calibration and application of the branched GDGT temperature proxy on East African lake sediments. *Earth and Planetary Science Letters*, *357*, 277-288.

McManus, J., Collier, R. W., Chen, C. T. A., and Dymond, J. (1992). Physical properties of Crater Lake, Oregon: A method for the determination of a conductivity-and temperature-dependent expression for salinity. *Limnology and Oceanography*, *37*(1), 41-53.

Pearman, J. K., Biessy, L., Thomson-Laing, G., Waters, S., Vandergoes, M. J., Howarth, J. D., Rees, A., Moy, C., Pochon, A., and Wood, S. A. (2020). Local factors drive bacterial and microeukaryotic community composition in lake surface sediment collected across an altitudinal gradient. FEMS Microbiology Ecology, 96(6), fiaa070.

Ramos-Roman, M. J., De Jonge, C., Magyari, E., Veres, D., Ilvonen, L., Develle, A. L., and Seppä, H. (2022). Lipid biomarker (brGDGT)-and pollen-based reconstruction of temperature change during the Middle to Late Holocene transition in the Carpathians. *Global and Planetary Change*, *215*, 103859.

Watson, B. I., Williams, J. W., Russell, J. M., Jackson, S. T., Shane, L., and Lowell, T. V. (2018). Temperature variations in the southern Great Lakes during the last deglaciation: Comparison between pollen and GDGT proxies. *Quaternary Science Reviews*, *182*, 78-92.

Wu, J., Yang, H., Pancost, R. D., Naafs, B. D. A., Qian, S., Dang, X., Sun, H., Pei, H., Wang, R., Zhao, S., and Xie, S. (2021). Variations in dissolved O2 in a Chinese lake drive changes in microbial communities and impact sedimentary GDGT distributions. *Chemical Geology*, *579*, 120348.

Zhang, Z., Smittenberg, R. H., and Bradley, R. S. (2016). GDGT distribution in a stratified lake and implications for the application of TEX86 in paleoenvironmental reconstructions. *Scientific reports*, *6*(1), 34465.